# Hypo3D: Exploring Hypothetical Reasoning in 3D

Ye Mao [1]   Weixun Luo [1]   Junpeng Jing [1]   Anlan Qiu [1]   Krystian Mikolajczyk [1]

## Abstract

The rise of vision-language foundation models marks an advancement in bridging the gap between human and machine capabilities in 3D scene reasoning. Existing 3D reasoning benchmarks assume real-time scene accessibility, which is impractical due to the high cost of frequent scene updates. To this end, we introduce *Hypothetical 3D Reasoning*, namely Hypo3D, a benchmark designed to evaluate models' ability to reason without access to real-time scene data. Models need to imagine the scene state based on a provided change description before reasoning. Hypo3D is formulated as a 3D Visual Question Answering (VQA) benchmark, comprising 7,727 context changes across 700 indoor scenes, resulting in 14,885 question-answer pairs. An anchor-based world frame is established for all scenes, ensuring consistent reference to a global frame for directional terms in context changes and QAs. Extensive experiments show that state-of-the-art foundation models struggle to reason effectively in hypothetically changed scenes. This reveals a substantial performance gap compared to humans, particularly in scenarios involving movement changes and directional reasoning. Even when the change is irrelevant to the question, models often incorrectly adjust their answers. The code and dataset are publicly available at: https://matchlab-imperial.github.io/Hypo3D.

## 1. Introduction

> *"Imagination is more important than knowledge."*
> – ALBERT EINSTEIN

Artificial General Intelligence (AGI) aims to replicate the

[1] Department of Electrical and Electronic Engineering, Imperial College London, United Kingdom. Correspondence to: Junpeng Jing <j.jing23@imperial.ac.uk>.

*Proceedings of the 42nd International Conference on Machine Learning*, Vancouver, Canada. PMLR 267, 2025. Copyright 2025 by the author(s).

full spectrum of human cognitive abilities (Goertzel, 2014; Rayhan et al., 2023). Reasoning, a core cognitive ability, is a major research focus. The emergence of vision-language foundation models (Hong et al., 2023; Liu et al., 2024; Wang et al., 2024; Fu et al., 2024; Mao et al., 2024) has marked a significant step toward narrowing the gap between human and machine reasoning, enabling progress from text summarization (Lewis, 2019; Zhang et al., 2020) to complex scene understanding (Huang et al., 2023; Driess et al., 2023).

These models are typically evaluated under the assumption that all ***knowledge*** about perception is immediately accessible during reasoning. For instance, in the evaluation of model performance in 3D scene reasoning, existing benchmarks (Ye et al., 2021; Azuma et al., 2022; Ma et al., 2022; Linghu et al., 2024; Zhang et al., 2024) presuppose real-time availability of scene data (e.g., point clouds). Yet, this assumption does not always hold, as real-world scenes are dynamic, and maintaining up-to-date 3D scenes is challenging. Unlike 2D image capture, 3D scene collection demands specialized equipment, extended scanning times, and a complex reconstruction process for accurate geometric representation (Nießner et al., 2013; Daneshmand et al., 2018). Thus, immediate perceptual knowledge for reasoning is unavailable in many real-world cases. ***Imagination***, rooted in prior knowledge, allows humans to overcome such limitations by deducing missing details and approximating reality. Even without direct visual input, humans can mentally simulate changes in a scene and reason about the changed scene in their minds. This ability, known as mental imagery (Pylyshyn, 2002; Moulton & Kosslyn, 2009), is crucial to human intelligence and raises a critical question: *Can current foundation models employ imagination to fill perceptual knowledge gaps and enhance reasoning?*

In response to this question, we propose a new reasoning concept: hypothetical reasoning. It evaluates models' ability to reason without immediate perceptual knowledge, requiring models to formulate reasonable hypotheses to bridge knowledge gaps. As an initial attempt, this study focuses specifically on hypothetical reasoning in 3D scenes, referred to as Hypo3D. As shown in Figures 1 and 2, the Hypo3D task follows a 3D visual question-answering format but cannot be solved using the given scene alone. Instead, models need to rely on an accessible context change description to imagine the current scene state after the change and adjust

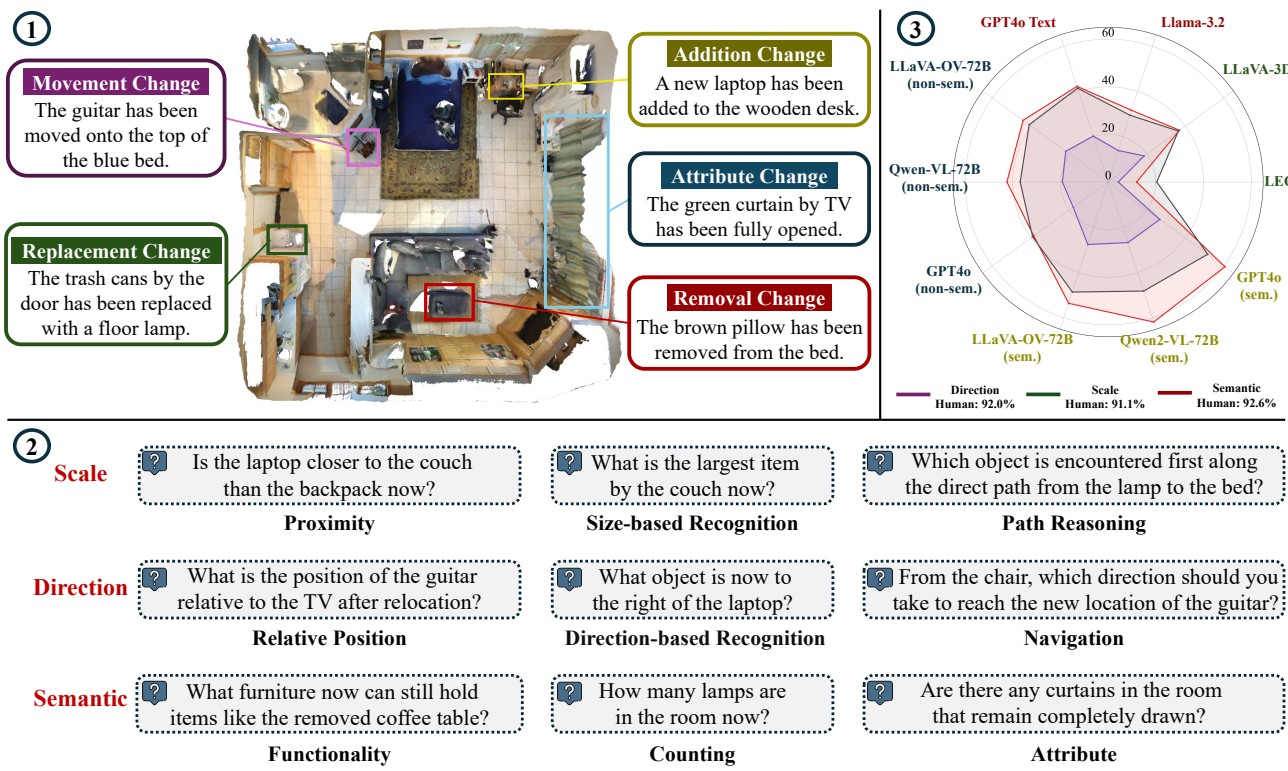

Figure 1. Overview of the Hypo3D benchmark. ① Examples of five context change types. ② Sample questions, including scale-based and direction-based questions requiring spatial reasoning, as well as semantic questions, all of which have open-ended answers. ③ The radar chart highlights a notable performance gap between models and humans, especially in direction-based questions.

their answers accordingly. Based on the Hypo3D task, we constructed a dataset comprising 7,727 context changes and 14,885 question-answer pairs across 700 indoor scenes. As shown in Figure 1, these context changes span five categories: (1) Movement Change, involving geometric transformations like translation or rotation; (2) Removal Change, taking away objects; (3) Attribute Change, modifying object properties such as color and state; (4) Addition Change, introducing new objects; and (5) Replacement Change, substituting existing objects with new ones.

The questions in the dataset range from simple proximity and relative position queries to intricate path reasoning and navigation tasks, broadly categorized into three types: (1) scale-based, focusing on proximity and size relationships; (2) direction-based, requiring reasoning about directional terms; and (3) semantic, highlighting scene semantics with minimal spatial reasoning. In constructing direction-based questions, we observe that existing 3D reasoning datasets reveal ambiguities in defining directional terms in 3D. Early datasets (Ye et al., 2021; Azuma et al., 2022) employ object-centric definitions, which lead to confusion when dealing with symmetrical or amorphous objects, such as round tables or cushions. Recent datasets (Ma et al., 2022; Linghu et al., 2024; Zhang et al., 2024) define directions relative to the observer. This strategy struggles to describe global di-

rectional relationships and introduces inconsistencies when multiple observers are involved. Towards this, Hypo3D establishes a world frame anchored to reference objects in each scene, ensuring that all directional terms are defined relative to a consistent global reference frame.

Extensive experiments on ten foundation models reveal a substantial performance gap between models and humans on the Hypo3D task, particularly in movement changes and directional reasoning. Surprisingly, closed-source models (e.g., GPT-4o (OpenAI, 2024)) do not outperform open-source alternatives. Furthermore, the models frequently hallucinate context changes, modifying their answers even when those changes are irrelevant to the posed question. Notably, their performance consistently degrades when required to imagine a scene after changes before reasoning, as compared to directly reasoning with the provided scene. These findings highlight a key limitation of current models: the inability to leverage imagination to infer missing perceptual knowledge and reason hypothetically.

## 2. Related Work

### 2.1. 3D Scene Update

The evolving nature of 3D environments requires continuous scene updates for effective understanding, particularly

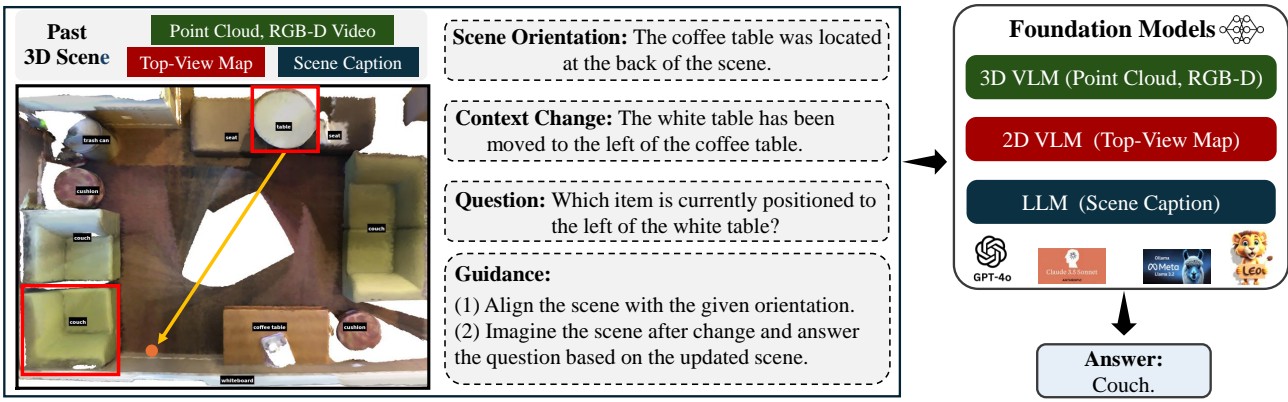

*Figure 2.* Example of hypothetical reasoning in a 3D scene. Given a 3D scene and an anchor-based frame description (Scene Orientation), models first align the scene to the specified frame. Then, based on a context change description and a question, models hypothetically modify the aligned scene and answer questions about the changed scene. Various models (e.g., LLMs, 2D VLMs, 3D VLMs) can tackle this task using corresponding scene representations, including scene captions, top-view maps, point clouds, and egocentric RGB-D videos.

in autonomous driving (Yurtsever et al., 2020) and robotic navigation (Wong & Spetsakis, 2000). A fundamental approach to scene updating involves reconstructing the entire scene from scratch. Advances in 3D reconstruction, such as multi-view stereo (Seitz et al., 2006; Jing et al., 2025), depth sensor-based (Zollhöfer et al., 2014), and volumetric methods (Newcombe et al., 2011), have improved reconstruction fidelity. But current methods struggle to balance accuracy, efficiency, and scalability (Nießner et al., 2013), making reconstruction primarily suitable for situations involving major changes. Recently, radiance-based 3D scene editing approaches, such as NeRF and 3D Gaussian Splatting (3DGS), have emerged as more efficient alternatives for incremental scene updates. NeRF-based methods (Liu et al., 2021; Kania et al., 2022) were limited to basic object edits and struggled with complex, cluttered scenes (Ye et al., 2025). 3DGS-based methods enable finer control over scene content, including geometry (Huang et al., 2024; Waczyńska et al., 2024; Ye et al., 2025), texture (Chen et al., 2024), and lighting (Gao et al., 2025). However, they struggle to disentangle these components and require costly re-optimization, limiting editability and efficiency (Wu et al., 2024).

Therefore, accurate and efficient 3D scene updates remain a challenge, making real-time scene acquisition not always feasible. This highlights the need for hypothetical reasoning.

### 2.2. 3D Visual Question Answering

3D Visual Question Answering (3D VQA), a key task for evaluating 3D reasoning, has advanced with the rise of Vision-Language Models (VLMs) (Hong et al., 2023; Liu et al., 2024; Anthropic, 2024). By integrating vision encoders with Large Language Models (LLMs) (Peng et al., 2023; Bai et al., 2023), VLMs enable multimodal perception of 3D scenes using inputs like top-view maps and point

clouds. As models advance, there is growing interest in developing more comprehensive benchmarks to better assess their 3D reasoning capabilities. Initial 3DQA dataset (Ye et al., 2021), derived from ScanNet (Dai et al., 2017), introduced 6K manually annotated QA pairs for scene-level reasoning. ScanQA (Azuma et al., 2022) expanded this with 41K QA pairs using automated question generation and human refinement. Qian et al. (Qian et al., 2024) further extended 3D VQA to autonomous driving scenarios, offering domain-specific challenges. SQA3D (Ma et al., 2022) introduced "situated reasoning," requiring models to contextualize answers based on an agent's position and orientation. MRSA (Linghu et al., 2024) and SPARTUN3D (Zhang et al., 2024) further scaled SQA3D, adding multimodal inputs such as images for richer situational context.

Unlike previous benchmarks, where answers are derived fully from the given scene, Hypo3D hypothetically applies a change to the scene and derives answers from the changed scene, increasing the hallucination risk.

## 3. Hypo3D Benchmark

This section first defines the Hypo3D task and explains how humans, assisted by LLMs, generate high-quality context changes and QAs. It then presents the dataset statistics.

### 3.1. Task Definition

A task instance in Hypo3D is formulated as a tuple $\langle S, F, c, q \rangle$, where $S$ denotes the scene representation; $F$ specifies the world frame defined by anchor objects in $S$ that standardizes all directional terms in the task; c describes a context change to be applied to the scene; and q denotes a question. The task is to first rotate $S$ into $\tilde{S}$ so that the anchor object is located in the orientation described in $F$, followed by computing an

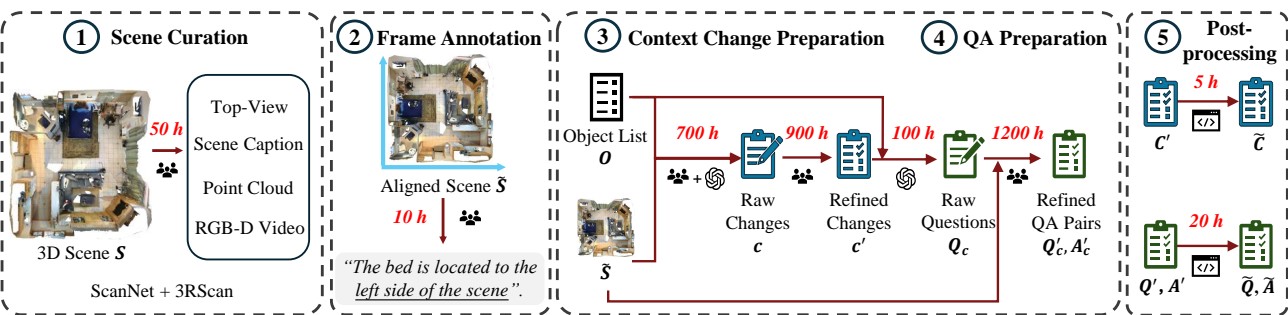

*Figure 3.* Dataset Generation Pipeline. The Hypo3D collection pipeline consists of five stages: Stage ① curates scenes (50 hours per person), Stage ② defines world frames (10 h/p), Stages ③ and ④ collect context changes and QA descriptions from human annotators (thousands of hours) and LLM, and Stage ⑤ conducts grammar checks and filters data based on semantic similarity. (25 h/p).

answer $a$ to $q$ after applying $c$ to $\tilde{S}$. Figure 2 provides an example of hypothetical reasoning in a 3D scene.

### 3.2. Dataset Generation Pipeline

As depicted in Figure 3, the Hypo3D benchmark was constructed through a rigorously designed five-stage pipeline. The initial Stage ① involves a comprehensive 50-hour manual curation process to collect and standardize diverse 3D scene representations from established datasets. This is followed by Stage ②, a meticulous 10-hour manual annotation phase where each scene is assigned an anchor-based world frame to ensure spatial consistency. The core data generation occurs in Stages ③ and ④, where we implemented a hybrid approach combining human expertise with LLMs to generate context changes and question-answer pairs. These critical stages demanded substantial human oversight, with over 1,000 hours dedicated to human refinement to guarantee exceptional data quality and contextual coherence. Finally, the pipeline concludes with Stage ⑤, a 25-hour post-processing phase that employs quality control measures to eliminate redundancies while maintaining grammatical precision.

**Scene Curation.** Each 3D scan $S$ in the Hypo3D benchmark can be represented through multiple modalities to facilitate reasoning, including point cloud, RGB-D video, top-view map, and scene caption. The point clouds and RGB-D videos are directly curated from the original scene datasets. Two variants of top-view maps are generated: non-semantic and semantic. Non-semantic maps are generated by positioning a simulated orthographic camera above the scene to produce photorealistic top-down renderings. Semantic maps augment non-semantic ones by incorporating hoverable semantic labels, with each label precisely positioned at the centroid of the bounding box corresponding to its object. An example semantic map can be seen in Figure 2. Unlike their non-semantic counterparts, semantic maps reduce object recognition errors in models, prioritizing the

evaluation of hypothetical reasoning performance. Scene captions are textual descriptions detailing the attributes of objects and their relationships within the scenes.

**Anchor-Based World Frame Annotation.** The same scene in 3D can be captured from different viewpoints in any orientation. Thus, it does not have a clearly defined, fixed reference frame. Here, we defined the world coordinate frame textually based on the locations of anchor objects in each scene, such as: "*The desk is located to the left of the scene*". When selecting anchor object candidates, visual prominence is prioritized over the extent to which they cover multiple sides of the scene. This is because, conventionally, defining a single primary orientation (e.g., "front") is sufficient to infer the remaining orientations ("left", "right", "back") using the right-hand rule (Hamilton, 2008), ensuring a consistent and unambiguous spatial frame.

**Context Change Preparation.** The Hypo3D benchmark defines five distinct types of object-level context changes, illustrated in Figure 1. For a scene $S$ with an object list $O = \{o_1, o_2, \ldots, o_n\}$, each change type is defined as follows:

(1) ***Movement changes*** relocate objects $O_m \subseteq O$ to new positions or orientations within the scene $S$.
(2) ***Removal changes*** eliminate objects $O_r \subseteq O$ from $S$, resulting in an updated object list $\hat{O} = O \setminus O_r$.
(3) ***Attribute changes*** modify properties of a subset of objects $O_{att} \subseteq O$, such as color, material, or state.
(4) ***Addition changes*** introduce new objects $O_a$ into $S$, updating the object list to $\hat{O} = O \cup O_a$.
(5) ***Replacement changes*** replace objects $O_{rp} \subseteq O$ with new objects $O_a$, resulting in an updated object list $\hat{O} = (O \setminus O_{rp}) \cup O_a$.

As illustrated in Figure 3 ③, human annotators and GPT-4o initially generate raw context changes uniformly to ensure data diversity. Human data is collected via crowdsourcing

on the CloudResearch platform.[1] For each scene aligned to the world frame $\tilde{S}$ with an associated object list $O$, annotators create a set of distinct descriptions $C$ for a specified change category, following the rules below:

(1) Each object $o_i \in O$ referenced in change $c_i \in C$ must have a uniquely specified location if it appears multiple times in the scene $\tilde{S}$.
(2) Each context change $c_i$ must be spatially feasible within the layout of $\tilde{S}$.
(3) Each $c_i$ must be independent, meaning it is derived solely from the original scene $\tilde{S}$ and does not rely on any version of $\tilde{S}$ modified by $c_j$, where $i \neq j$.

For GPT-4o, context changes are generated using the semantic top-view map of $\tilde{S}$ and a textual prompt that follows the same criteria as those provided to human annotators. The detailed prompts and human guidelines for raw change generation are provided in Appendix A.

Finally, each raw change $c$ generated by humans and GPT-4o is edited by an independent group of human reviewers, producing a refined version $c'$.

**Question-Answer Preparation.** Raw questions are initially generated by GPT-4o, as illustrated in Figure 3 ④. Seven prompt templates, each corresponding to a specific question type, are designed for each type of context change. Each template includes the object list $O$ in the scene $S$, the context change $c$, example questions, and a question type description (e.g., ask for the position of the changed object relative to other objects in the scene). In total, eleven unique question types are evenly distributed across the context change types (see Appendix A.2). For each context change $c$, 21 raw questions are generated, denoted as $Q_c$.

Human reviewers then refine $Q_c$ to produce a new question set $Q'_c$. During this process, only a small portion of raw questions are retained. Specifically, 91% of the questions are filtered out, and the remaining questions undergo additional editing to strictly ensure $Q'_c$ satisfies the following criteria: (1) each $q \in Q'_c$ can only be answered by combining $S$ and $c$, as neither is sufficient on its own; (2) the answer to each $q$ must be potentially impacted by the change $c$; (3) each $q$ has a unique and unambiguous answer; and (4) answers cannot be inferred from commonsense knowledge (e.g., a bed is larger than a pillow). Finally, all questions in $Q'_c$ are annotated with concise human-provided answers $A'_c$ and reclassified into general types to evaluate models based on their reasoning capabilities: scale-based, direction-based, and semantic questions, as shown in Figure 1. Scale-based questions assess spatial reasoning related to proximity and size perception, direction-based questions focus on orientation understanding, and semantic questions evaluate the

model's ability to interpret object attributes with minimal spatial reasoning. Notably, questions can belong to multiple types if they require diverse reasoning.

**Post-processing.** To ensure data diversity, SBERT (Reimers, 2019) is used to remove semantically similar descriptions. It encodes the refined context changes $C'$ and questions $Q'$, producing embeddings $E_{C'}$ and $E_{Q'}$, respectively. Context changes and questions are filtered by excluding pairs with cosine similarity $\text{Sim}(e_i, e_j) > 0.8$ for $e_i, e_j \in E_{C'}$, and $\text{Sim}(e_k, e_v) > 0.8$ for $e_k, e_v \in E_{Q'}$. The remaining context changes, questions, and corresponding answers are further refined for grammatical accuracy using GPT4-Turbo, resulting in descriptions for the final dataset $\tilde{C}$, $\tilde{Q}$, and $\tilde{A}$.

### 3.3. Statistics

The Hypo3D benchmark comprises 700 unique scenes, with 500 sourced from the ScanNet (Dai et al., 2017) dataset and 200 from the 3RScan (Wald et al., 2019) dataset, randomly sampled from their respective sources. The dataset includes 7,727 context changes and 14,885 question-answer pairs. On average, a context change description contains 13.62 words, a question description contains 13.69 words, and an answer contains 1.28 words. The word cloud in Figure 4 ① highlights that the most frequent words in context change descriptions are verbs representing change actions, such as "moved," "removed," and "replaced". Figure 4 ② illustrates the distribution of context change types, with movement changes being the most frequent, as they often result in more pronounced scene layout rearrangements. The bar charts in Figure 4 ③ show that scale-based and direction-based questions, which require more spatial reasoning, constitute a larger portion of the dataset compared to semantic questions.

## 4. Experiments

### 4.1. Evaluation Protocol

*Exact Match (EM)* and *Partial Match (PM)* are the metrics used for evaluation. EM measures the percentage of model-predicted answers that exactly match the ground-truth answers. PM quantifies the percentage of overlapping words between the predicted answers and the ground truth. For the same model, PM is typically higher than EM, as it accounts for partial correctness. When computing EM and PM, both the predicted and ground-truth answers are normalized by lowercasing, removing punctuation, and aligning semantically similar terms (e.g., "left" and "west").

### 4.2. Baselines

A total of ten foundation models were evaluated in a zero-shot setting on the Hypo3D benchmark, grouped into: (1) LLMs, (2) 2D VLMs, and (3) 3D VLMs. The evaluated

---

[1]CloudResearch, Connect Platform, https://www.cloudresearch.com.

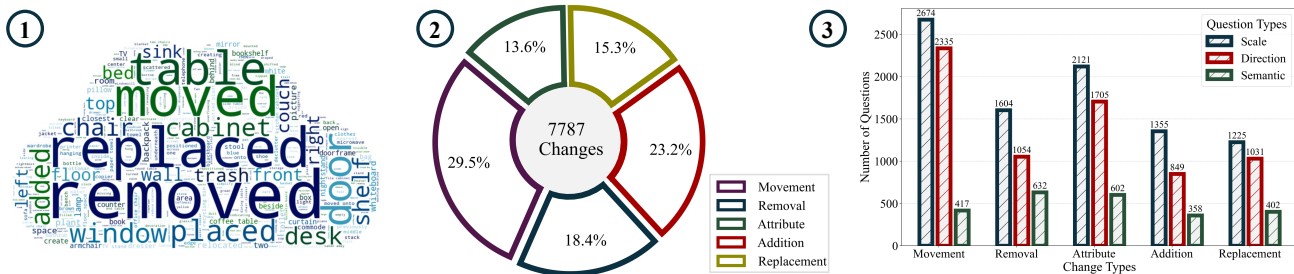

*Figure 4.* Dataset Statistics. ① Word cloud representing context change descriptions. ② Frequency distribution of context change types across 7,727 instances. ③ Distribution of question types across change categories, with question frequency consistently highest for scale-based, then direction-based, and finally semantic.

LLMs, Llama3.2 (3B) (Dubey et al., 2024) and GPT-4o (OpenAI, 2024), utilize textual scene captions from the ScanRefer (Chen et al., 2020) and SceneVerse (Jia et al., 2025) datasets to represent 3D scenes. The 2D VLMs consist of open-source models Qwen2-VL (7B, 72B) (Wang et al., 2024) and LLaVA-OV (7B, 72B) (Li et al., 2024a), alongside closed-source models GPT-4o (OpenAI, 2024) and Claude 3.5 Sonnet(Anthropic, 2024)[2], both employing semantic and non-semantic top-view maps. The 3D VLMs assessed include LEO (7B) (Huang et al., 2023), which encodes 3D scenes using egocentric 2D images with 3D point clouds, and LLaVA-3D (7B) (Zhu et al., 2024), which represents 3D scenes through multi-view images. All 3D scene point clouds have been explicitly aligned to a top-view perspective with the floor on the XY-plane and vertical structures along the Z-axis.

In addition to scene inputs, all models receive the anchor-based world frame, hypothetical context changes applied to the scenes, the question, and task guidance as part of the textual prompt. Specific prompt templates for each model are provided in Appendix B.2. As shown in Figure 2, The task guidance generally consists of three steps: (1) Rotate the scene to align with the provided world frame, (2) Imagine the scene after the specified context change, and (3) Answer the question based on the modified scene.

### 4.3. Results and Discussions

**Human-Model Performance Gap.** To manage the high costs of human evaluation, we sampled 50 scenes and 250 context changes with 50 questions per change type for assessment. To avoid contamination, human evaluators were excluded from benchmark annotation and provided only 10 QA pairs for task familiarization. As shown in Table 1, human performance exceeds 85% in EM across all change types, though it doesn't reach 100% due to the open-ended nature of Hypo3D questions. Most errors from human evaluators were due to typos, vague phrasing, formatting mis-

matches, and inherent noise in 3D scenes. Human performance is slightly lower for addition and replacement changes, as these introduce new objects, causing confusion with existing ones.

In contrast, foundation models perform significantly worse, with the top-performing model, GPT-4o, achieving under 50% overall in both EM and PM metrics, even with a semantic top-view map input. Notably, it lags 45.5% behind human EM performance. Furthermore, models show significant performance bias across different question types, as highlighted by the distinct non-overlapping regions in the radar chart in Figure 5. In comparison, humans achieve over 90% accuracy across all question types, demonstrating consistently strong performance.

**LLMs *vs.* 2D VLMs *vs.* 3D VLMs.** Table 1 shows that 2D VLMs outperform other foundation models across all change types. Their performance improves significantly with semantic top-view maps that provide explicit object labels compared to non-semantic maps. When using non-semantic maps, most 2D VLMs perform worse than the text-only version of GPT-4o, highlighting the impact of image recognition errors on 2D VLM performance.

For open-source VLMs like Qwen2-VL and LLaVA-OV, larger model sizes yield better results. Closed-source models (GPT-4o and Claude 3.5 Sonnet), despite their reputation for superior performance, do not maintain this advantage on the Hypo3D task. They excel with semantic maps but struggle on non-semantic maps, where the open-source LLaVA-OV 72B delivers the best performance. This pattern aligns with findings by Li et al. (2024b), where closed-source models particularly struggled on unlabeled top-view maps.

Interestingly, 3D VLMs, despite receiving the richest geometric information, do not demonstrate a clear advantage over 2D VLMs or LLMs. Notably, the LEO model struggles with the instruction following, often failing to interpret task guidance and achieving the lowest EM score (14.83%). LLaVA-3D, although outperforming all other 7B 2D VLMs when using a non-semantic map, lacks a larger model size

---

[2]We use GPT-4o-08-16 and Claude 3.5 Sonnet-10-22.

*Table 1.* EM and PM accuracy of ten foundation models and human evaluators on Hypo3D. The highest model performance for each type of context change is in **bold**, while the best-performing model within each family is underlined.

| Model | Movement | | Removal | | Attribute | | Addition | | Replacement | | Overall | |
|---|---|---|---|---|---|---|---|---|---|---|---|---|
| | EM | PM | EM | PM | EM | PM | EM | PM | EM | PM | EM | PM |
| *LLM (Scene Caption)* | | | | | | | | | | | | |
| Llama-3.2 3B | 25.31 | 28.37 | 29.85 | 33.65 | 24.95 | 29.59 | 26.78 | 30.78 | 23.75 | 27.68 | 26.08 | 29.91 |
| GPT-4o API (Text) | 35.76 | 38.66 | 36.88 | 41.71 | 34.05 | 39.58 | 39.74 | 43.28 | 31.33 | 35.24 | 35.54 | 39.65 |
| *2D VLM (Non-Semantic Top-View Map)* | | | | | | | | | | | | |
| Qwen2-VL 7B | 29.23 | 35.08 | 30.71 | 34.69 | 29.04 | 33.94 | 31.48 | 35.17 | 28.41 | 33.10 | 29.68 | 34.47 |
| Qwen2-VL 72B | 33.02 | 37.38 | 33.88 | 37.57 | 33.48 | 37.62 | 35.95 | 40.29 | 30.66 | 34.64 | 33.39 | 37.51 |
| LLaVA-OV 7B | 30.34 | 34.17 | 29.81 | 33.24 | 31.37 | 36.13 | 33.12 | 35.64 | 28.41 | 31.81 | 30.62 | 34.34 |
| LLaVA-OV 72B | 36.46 | 39.83 | 36.45 | 40.22 | 35.70 | 40.46 | 39.64 | 42.25 | 33.83 | 37.85 | 36.38 | 40.13 |
| Claude 3.5 Sonnet API | 17.49 | 30.24 | 19.90 | 27.34 | 22.96 | 33.47 | 22.90 | 31.61 | 20.35 | 27.70 | 20.42 | 30.29 |
| GPT-4o API | 34.49 | 37.69 | 32.85 | 36.53 | 31.23 | 35.38 | 38.09 | 40.70 | 30.04 | 33.22 | 33.58 | 36.75 |
| *2D VLM (Semantic Top-View Map)* | | | | | | | | | | | | |
| Qwen2-VL 7B | 31.26 | 36.41 | 38.09 | 41.90 | 34.83 | 39.41 | 37.64 | 41.41 | 31.86 | 36.62 | 34.40 | 38.91 |
| Qwen2-VL 72B | 38.42 | 42.56 | **47.36** | **51.05** | 46.76 | 51.10 | 47.63 | 50.87 | **44.43** | 48.78 | 44.25 | 48.25 |
| LLaVA-OV 7B | 33.32 | 36.80 | 34.34 | 37.84 | 34.98 | 39.50 | 38.96 | 41.98 | 33.93 | 38.33 | 34.81 | 38.60 |
| LLaVA-OV 72B | 39.39 | 42.99 | 43.44 | 46.87 | 44.57 | 49.37 | 46.12 | 49.06 | 44.10 | 48.18 | 43.01 | 46.83 |
| Claude 3.5 Sonnet API | 30.92 | 42.98 | 40.26 | 48.54 | 42.29 | **52.72** | 43.16 | 51.59 | 43.28 | **50.73** | 38.86 | 48.65 |
| GPT-4o API | **40.77** | **43.79** | 47.36 | 50.40 | **47.42** | 51.39 | **50.59** | **53.77** | 44.24 | 47.68 | **45.50** | **48.82** |
| *3D VLM (RGB-D Video, Point Cloud)* | | | | | | | | | | | | |
| LEO 7B | 14.40 | 22.96 | 18.54 | 22.82 | 14.35 | 21.56 | 14.64 | 24.83 | 11.76 | 19.50 | 14.83 | 22.40 |
| LLaVA-3D 7B | 31.63 | 35.11 | 30.60 | 33.91 | 31.60 | 36.16 | 33.67 | 36.70 | 30.42 | 34.16 | 31.56 | 35.23 |
| **Human** | 95.00 | 96.00 | 93.00 | 95.00 | 93.00 | 94.83 | 89.00 | 90.67 | 85.00 | 86.00 | 91.00 | 92.50 |

variant to fully showcase its potential in this task. See Appendix B.4 for additional qualitative results illustrating the performance gap between models.

## 4.4. Analyses and Insights

Our key insights are as follows: Models face significant difficulties in reasoning about hypothetically changed scenes compared to static ones, particularly with movement and replacement changes. Direction-based questions that require scene orientation also pose challenges, with performance deteriorating in the absence of a defined world frame. Most models exhibit severe hallucinations, often altering their answers even when the context changes are irrelevant to the questions. All 2D VLM results presented in Tables 2, 3, and 4 are derived from the semantic top-view map.

**Insight 1**: *Models struggle with hypothetical movement and replacement changes.*

The bolded values in Table 1 indicate the highest EM and PM accuracy achieved by models across different context changes. Movement changes show the lowest performance, scoring 9.82% lower in EM and 9.98% lower in PM compared to the best-performing addition changes. This outcome highlights the models' difficulty in handling changes that heavily reconfigure the scene's spatial layout and alter

inter-object spatial relationships. Another finding is that replacement changes perform worse than both removal and addition changes, likely because they involve both object removal and addition simultaneously, making them more challenging than handling either change individually. For example, the replacement change "*a cup is replaced with a phone*" can be interpreted as first removing the cup and then adding the phone in its place.

**Insight 2**: *Models struggle with direction-based questions.*

The radar chart in Figure 5 shows EM results across different question types, revealing that most models, except LEO, perform better on semantic questions than spatial ones. A more distinct pattern can be observed in Figure 14. When using a semantic map where object labels are provided, leading models like Qwen2-VL 72B and GPT-4o achieve over 60% EM, indicating that semantic questions are less challenging when models correctly identify objects. Within spatial questions, models struggle more with direction-based questions compared to scale-based ones. Even with scene input aligned to the world frame, performance on direction-based questions remains low (see Appendix B.3.2), suggesting models struggle more with orientation understanding than with size and proximity reasoning.

**Insight 3**: *Anchor-based frame definition improves orienta-*

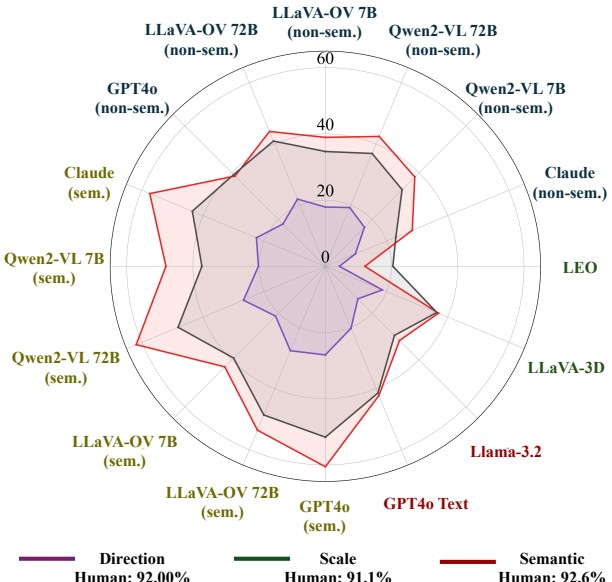

Figure 5. Model and human EM performance across question types. Humans consistently achieve strong performance, whereas models struggle, particularly with direction-based questions.

Table 2. Comparison of model performance on directional questions using no frame, camera view frame, and anchor object frame. Red is lower, and green is higher, compared to w/o frame.

| Model | w/o frame | | w. camera | | w. anchor (ours) | |
|---|---|---|---|---|---|---|
| | EM | PM | EM | PM | EM | PM |
| Llama 3.2 3B | 9.7 | 18.55 | - | - | 13.36 | 21.52 |
| Qwen2-VL 72B | 20.54 | 30.61 | 19.97 | 30.14 | 22.03 | 33.63 |
| GPT-4o API | 18.45 | 28.36 | 18.20 | 28.04 | 19.37 | 29.48 |
| LLaVA-3D 7B | 15.57 | 25.57 | - | - | 15.31 | 25.78 |

*tion understanding.*

Table 2 presents model results on 3,495 pure direction-based questions from Hypo3D under three conditions: without a world frame (w/o frame), using the camera view as the frame (w. camera), and using the anchor-based frame (w. anchor). The camera view is an image that captures the $X$ side of the room from its center, where $X \in \{$left, right, front, back$\}$. The results indicate that not all frame definition methods can be effectively interpreted by current models. Only our anchor-based definition method consistently improves the performance of 2D and 3D VLMs, whereas using the camera view as a frame reduces the performance. See Appendix B.2 for details on inference templates across different settings.

**Insight 4**: *Reasoning in hypothetically changed scenes is more challenging than in unchanged scenes.*

250 context changes and corresponding question pairs were sampled from Hypo3D to validate this insight. Each pair was annotated with two answers: one based on the unchanged scene and the other based on the hypothetically changed scene. Table 3 presents model performance for

Table 3. Comparison of model performance when using and not using context change, where the changes **affect** the answer.

| Model | w/o change | | w. change | |
|---|---|---|---|---|
| | EM | PM | EM | PM |
| LLaMA-3.2 3B | 19.00 | 23.25 | 20.50 (+1.50) | 24.50 (+1.25) |
| Qwen2-VL 72B | 37.00 | 41.50 | 31.50 (-5.50) | 36.00 (-5.50) |
| GPT-4o API | 38.00 | 40.25 | 33.00 (-5.00) | 36.00 (-4.25) |
| Claude 3.5 Sonnet API | 33.00 | 39.75 | 29.00 (-4.00) | 35.50 (-4.25) |
| LLaVA-3D 7B | 27.00 | 31.00 | 20.50 (-6.50) | 24.00 (-7.00) |

answering questions under both conditions. Most models, except Llama-3.2 3B, exhibit a consistent performance drop in EM and PM accuracy when reasoning in changed scenes compared to the unchanged scene. This finding addresses the core research question discussed in Sec. 1, showing that the imagination capability required for hypothetical reasoning is lacking in current foundation models.

Table 4. Comparison of model performance when using and not using context change, where the changes **do not affect** the answer.

| Model | w/o change | | w. change | |
|---|---|---|---|---|
| | EM | PM | EM | PM |
| LLaMA-3.2 3B | 27.50 | 31.42 | 29.00 (+1.50) | 33.25 (+1.83) |
| Qwen2-VL 72B | 56.50 | 60.17 | 51.50 (-5.00) | 55.17 (-5.00) |
| GPT-4o API | 57.00 | 60.00 | 52.50 (-4.50) | 56.92 (-3.08) |
| Claude 3.5 Sonnet API | 52.50 | 59.00 | 49.00 (-3.50) | 53.25 (-5.75) |
| LLaVA-3D 7B | 37.50 | 40.17 | 37.00 (-0.50) | 40.17 (0.00) |

**Insight 5**: *Models hallucinate when changes are irrelevant.*

Previous results indicate that models struggle to understand how context changes influence answers. Here, we show that they also struggle to ignore irrelevant context. To test this, we constructed 250 new context change-QA triplets, separate from the benchmark. In these triplets, the context changes have no impact on the answers to the questions. For example, the context change can be "*The object is moved from A to B*", while the question asks, "*What is the color of the object?*" Models were evaluated both with and without the context change description. Ideally, they should provide consistent answers, as object movement does not affect its color. However, as shown in Table 4, all 2D and 3D VLMs exhibit performance degradation when context changes are introduced. Notably, while 2D VLMs achieve the highest performance on Hypo3D tasks in previous evaluations, they also exhibit more severe hallucinations.

## 5. Conclusion

In this paper, we introduce the Hypo3D task to investigate the hypothetical reasoning ability of foundation models in 3D. This task challenges models to imagine scene changes before 3D reasoning, demanding robust reasoning without real-time scene access. To standardize directional term definitions in 3D, Hypo3D employs an anchor-based world

frame for each scene. Extensive experiments on ten foundation models, including LLMs, 2D VLMs, and 3D VLMs, reveal that all models struggle with the Hypo3D task, especially when handling movement changes and directional reasoning questions. These models exhibit severe hallucinations, altering answers even when the context changes do not affect the questions. These findings confirm that current models struggle to simulate scene changes without direct observation. We hope this study inspires further research into strengthening foundation models' hypothetical reasoning to narrow the gap with human cognitive abilities.

## Acknowledgments

We would like to thank Endong Sun, Aiden Deng, Wenqiang Lai, Yicheng Zhan, and Zihan Jiang for their assistance in pre-testing the website for human crowdsourcing. We are also grateful to Eva Yin, Yan Lin, Vasandani Gavin, and all annotators from the CloudResearch platform for their contributions in providing high-quality context changes and answers for the Hypo3D dataset. We thank Chengzu Li for his valuable feedback on early drafts of the paper, and Jiangnan Ye for his help in designing prompt templates for collecting context changes using LLMs. This research was supported by the Imperial College President's PhD Scholarship.

## Impact Statement

The impact of our study is multifaceted. First, while hypothetical reasoning is motivated by challenges in handling frequent updates in 3D scenes, it is crucial for foundation models across modalities, including 3D, 2D, and text. It enables models to simulate perceptual outcomes internally without requiring physical action. This capability is especially valuable in scenarios where obtaining sensory output is technologically constrained or costly. Second, Hypo3D, the first 3D VQA benchmark, standardizes directional relationships using a fixed world frame, enabling unified representations. Finally, we will release prompt templates, WebUI source code for data collection, 3D scene representations, and reasoning scripts for model evaluation, empowering researchers to develop customized hypothetical reasoning datasets and ensuring result reproducibility.

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

# Appendix

In the appendix, we will present more details about Hypo3D benchmark, more experimental results, and limitations and future work.

## A. Benchmark Details

### A.1. Human Collection via Crowdsourcing

Half of the raw context changes were collected using the crowdsourcing platform CloudResearch, following a multi-stage procedure facilitated by a dedicated WebUI designed for collecting specific types of changes. Annotators were recruited from English-speaking countries, including the United States, Australia, Canada, Ireland, New Zealand, and the United Kingdom, to ensure linguistic consistency. Annotators were compensated at a rate of $1.00 for every five change descriptions derived from 3D scenes. Figure 6 outlines the annotation protocol, specifying requirements for accurate descriptions of moved object locations and ensuring that changed objects are not repeated across submissions.

### A.2. LLM Collection

GPT-4o is integral to both the context change and question collection processes in Hypo3D. For context change collection, each type of change is generated using dedicated prompt templates. Figure 7 highlights the templates specifically designed for capturing addition changes.

For the question collection, 11 unique question types are designed, encompassing categories such as proximity, size-based recognition, and path reasoning, as outlined in Table 5. Figure 8 and 9 illustrate the prompt templates for generating questions related to proximity and relative positions, respectively. Each context change is associated with 7 specific question types, and the distribution of question types across different changes is detailed in Table 6. While some context changes share the same question types, they are provided with different prompt templates and example questions for diverse generations. Once collected, the questions are categorized into broader, coarse-grained categories based on the primary capability required to answer them, such as scale-based, direction-based, and semantic questions.

Data Collection Guidelines --Please Read                                                                                    ⌃

## Welcome!

Explore the given 3D scene visualization and describe **five** different ways to move objects within it.

**Consider the example scene below, possible movements can include:**

- The brown pillow, originally on the bed, has been moved to the gray couch.
- The desk, which was next to the white cabinet, is now positioned between the refrigerator and the two trash cans.

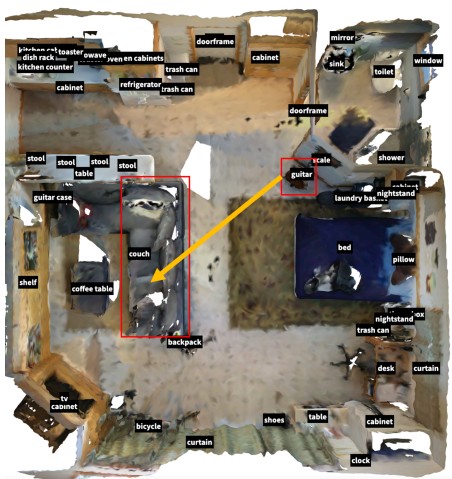

### Instructions:

- Movements must be spatially feasible within the scene's layout.
- Each movement description should clearly specify the object(s) being moved, its original location, and its new location in a unique way. Ambiguous or wrong descriptions will be **rejected**.

**Good Description:** The red apple on the table has been moved to the sink near to the refrigerator.
**Bad Description:** The apple has been moved to the sink. (Which apple? Which sink?)

- Each description should move different objects in the scene.
- Each description should be more than 10 words.
- All movements must occur within the same scene and be independent of one another.

Once you have finished the task, click the **Submit** button to receive your Completion Code.

*Please use your imagination and creativity to come up with unique and interesting movements!*

Figure 6. Guidelines for movement change collection in crowdsourcing.

*Table 5.* Comparison of model performance in direction-based questions on non-aligned versus aligned top-view maps. The results show no significant improvement, and in some cases, a decline in performance with aligned maps

| | |
|---|---|
| **Proximity** | *Which direct distance is shorter: from the nightstand to the window or to the box?* |
| **Direction-based Recognition** | *What item is directly positioned below the shelf now?* |
| **Size-based Recognition** | *What is the largest item remaining on the bed now?* |
| **Functionality** | *What item in the room now provides storage functionality similar to the removed shelf?* |
| **Counting** | *What is the current count of chairs next to the table?* |
| **Navigation** | *Which direction should you move from the plant to reach the new chair?* |
| **Placement Height** | *Is the box positioned higher or lower than the keyboards?* |
| **Relative Position** | *What is the relative position of the armchair to the footstool now?* |
| **Attribute** | *How many different colors of the table are in the room now?* |
| **Path Reasoning** | *Is the direct path from the trash can's new location to the fire alarm obstructed by any objects?* |
| **Situational Reasoning** | *Are you closer to the repositioned trash can or the copier when you're next to the door?* |

*Table 6.* Distribution of question types across different context changes.

| **Movement** | **Removal** | **Attribute** | **Addition** | **Replacement** |
|---|---|---|---|---|
| Proximity | Situational Reasoning | Proximity | Proximity | Proximity |
| Direction-based Recognition | Direction-based Recognition | Direction-based Recognition | Direction-based Recognition | Direction-based Recognition |
| Size-based Recognition | Size-based Recognition | Size-based Recognition | Size-based Recognition | Size-based Recognition |
| Relative Position | Size Fitness | Relative Position | Relative Position | Relative Position |
| Path Reasoning | Functionality | Functionality | Functionality | Functionality |
| Placement Height | Navigation | Navigation | Placement Height | Attribute |
| Counting | Counting | Counting | Counting | Counting |

---

## Addition Change Generation

Generate four unique object additions within the given 3D scene. Each change should clearly specify which object(s) are being added and provide detailed descriptions. Two changes should add more existing items, while two should introduce new items not present in the scene.

Requirements:
1. The added object's location and appearance should be described clearly.
2. Changes should be deterministic, detailed, realistic, and contextually appropriate for the scene.
3. The number of objects added in each change should be varied.

List the changes in the order 1 to 4, with each change starting on a new line.

---

*Figure 7.* Prompt template for addition change generation.

---

## Proximity Question Generation

*Input and Output*
The input consists of a 3D scene description, a context change description, and a question related to the change.

*Your Role*
Your task is to assist in generating questions for a 3D Visual Question Answering dataset. The dataset aims to evaluate the model's reasoning ability based on changes in a 3D scene.

*Guidelines for Generating Questions:*
- Focus each question on the proximity (close/far) between the added object and the other objects in the scene.
- The 'other objects' should not be the object mentioned in the change description.
- Use various starters such as "what," "where," "which" etc., to ensure diverse question structures.
- Make sure each question has a unique sentence structure and is phrased totally differently from the others.
- Avoid asking questions that can be inferred from the change. For example, if the change is, "A mug is added next to the laptop," avoid questions like, "What object is placed beside the laptop?".
- Each question should have a clear, definitive answer that eliminates ambiguity.

*Example:*
Object List: This scene contains 1 kitchen counter, 1 shower, 1 desk, 1 sink, 1 scale, 1 tv, 1 pillow, 1 clock, 1 backpack, 2 couch, 1 refrigerator, 1 coffee table, 1 toilet, 1 bed, 4 trash can.

Context Change: A new coffee machine has been added on the desk in the scene.

Example questions:
(1) Of the couch, refrigerator, and clock, which one is situated nearest to the added coffee machine?
(2) Between the couch and the clock, which item is farther away from the new coffee machine?

Object List for Question Generation{}
Context Changes for Question Generation{}

Now, generate three unique and diverse questions for each scene change above. The response should be in JSON format, with each change as a key and its questions as an array of values.

*Figure 8.* Prompt template for generating raw proximity questions based on context changes and the object list of the 3D scene.

## Relative Position Question Generation

*Input and Output*
The input consists of a 3D scene description, a context change description, and a question related to the change.

*Your Role*
Your task is to assist in generating questions for a 3D Visual Question Answering dataset. The dataset aims to evaluate the model's reasoning ability based on changes in a 3D scene.

*Guidelines for Generating Questions:*
- Focus on asking the relative position of other objects in the scene respect to the changed object. Potential answers could be (e.g., in front of, back, left, right).
- The 'other objects' should not be the object mentioned in the change description.
- Use various starters such as "what," "where," "which" etc., to ensure diverse question structures.
- Make sure each question has a unique sentence structure and is phrased totally differently from the others.
- Avoid asking questions that can be inferred from the change. For example, if the change is, "The brown pillow is moved to the front of the couch," avoid questions like, "What is behind the pillow?".
- Each question should have a clear, definitive answer that eliminates ambiguity.

*Example:*
Object List: This scene contains 1 kitchen counter, 1 shower, 1 desk, 1 sink, 1 scale, 1 tv, 1 pillow, 1 clock, 1 backpack, 2 couch, 1 refrigerator, 1 coffee table, 1 toilet, 1 bed, 4 trash can.

Context Change: The brown pillow, originally on the bed, has been moved to the gray couch.

Example questions:
(1) What is the current position of the coffee table compared to the pillow?
(2) How is the sink positioned relative to the relocated pillow now?

Object List for Question Generation{}
Context Changes for Question Generation{}

Now, generate three unique and diverse questions for each scene change above. The response should be in JSON format, with each change as a key and its questions as an array of values.

*Figure 9.* Prompt template for generating raw relative position questions based on context changes and the object list of the 3D scene.

*Table 7.* A comparison between Hypo3D and existing 3D vision-language datasets. "Hypothetical" indicates whether the dataset includes context changes. "Question Type" denotes whether questions are categorized into predefined types. "VG" and "QA" refer to visual grounding and question-answering, respectively. "World Frame" denotes if the scene's orientation in 3D space is explicitly defined.

| Dataset | Task | Question Type? | Hypothetical? | World Frame? | #Scans | #Language | Text Collection |
|---|---|---|---|---|---|---|---|
| ScanRefer (Chen et al., 2020) | VG | N/A | ✗ | ✗ | 0.7k | 11k | Human |
| Sr3D (Achlioptas et al., 2020) | VG | N/A | ✗ | ✗ | 0.7k | 115k | Template |
| ScanQA (Azuma et al., 2022) | QA | ✗ | ✗ | ✗ | 0.8k | 41k | Template |
| SQA3D (Ma et al., 2022) | QA | ✗ | ✗ | ✗ | 0.65k | 33.4k | Human |
| ScanScribe (Zhu et al., 2023) | Captioning | N/A | ✗ | ✗ | 1.2k | 278k | LLM |
| MMScan (Lyu et al., 2024) | VG + Captioning + QA | ✗ | ✗ | ✗ | 5.2k | 6.9M | LLM + Temp. + Human |
| **Hypo3D (Ours)** | QA | ✓ | ✓ | ✓ | 0.7k | 15k | LLM + Human |

## A.3. Comparison with Existing 3D Vision-Language Datasets

Comparisons with relevant 3D scene understanding tasks and benchmarks are summarized in Table 7.

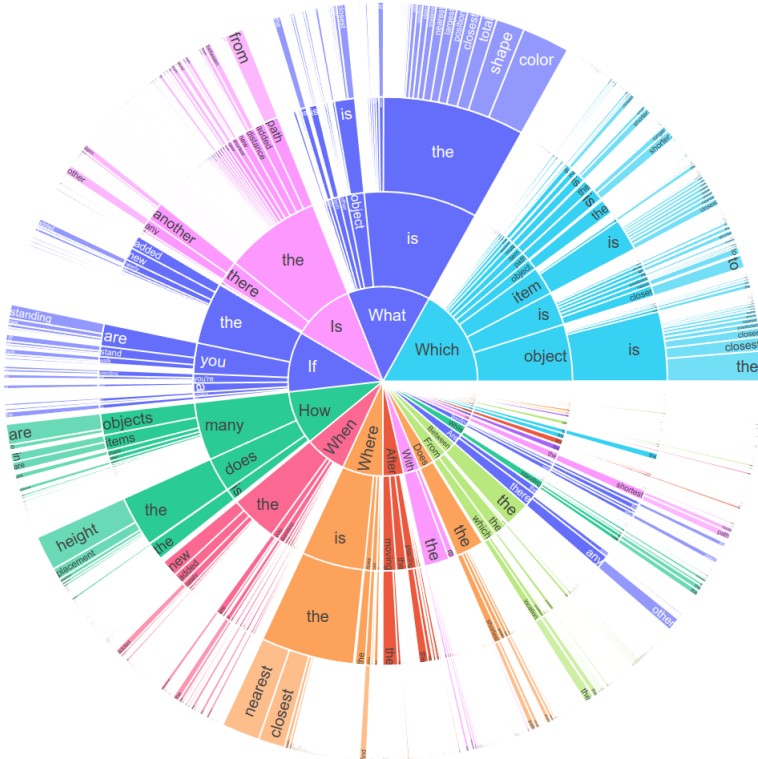

*Figure 10.* Question distribution in Hypo3D.

## A.4. Dataset Statistics

Figure 10 demonstrates that the questions in our benchmark begin with various start words, such as "what", "which", "is", "how", and "when". None of these words dominate the dataset, highlighting the balanced and diverse nature of our dataset.

# B. Experiments

## B.1. Model Hyperparameter Settings

Our experiments primarily used the default inference hyperparameters for zero-shot models, as detailed in Table 8. For Claude 3.5 Sonnet, the maximum new token parameter was set to 40, reduced from its default value due to the model's

tendency to generate lengthy responses, even when instructed to be concise.

*Table 8.* Inference hyperparameter settings for the baseline models.

| Model | Max New Tokens | Temperature |
|---|---|---|
| Llama3.2 3B | 32 | 0.6 |
| Qwen2-VL 7B & 72B | 128 | 0.01 |
| LLaVA-OV 7B & 72B | 128 | 0.7 |
| GPT-4o API | 1024 | 1.0 |
| Claude 3.5 Sonnet API | 40 | 1.0 |
| LLaVA-3D 7B | 512 | 0.2 |
| LEO 7B | 256 | 1.0 |

---

## Hypothetical Reasoning Prompt

*3D VLM:* Given a 3D scene, mentally rotate the scene to align with the specified orientation.

*2D VLM:* Given a top-view of a 3D scene, mentally rotate the image to align with the orientation.

*LLM:* Given a 3D scene description, mentally match it with the specified orientation.

Scene Orientation: {}

Now, given a context change, imagine how the scene would appear after the change has been applied. Then, answer a question based on the modified scene.

Context Change: {}
Question: {}

The answer should be a single word or a short phrase.

The answer is:

---

*Figure 11.* Prompt template for the main hypothetical reasoning experiments, with differences between prompts for 3D VLM, 2D VLM, and LLM underlined.

### B.2. Reasoning Prompts

The prompt template used for the main results in Table 1 is shown in Figure 11. The only difference between prompts for LLM, 2D VLM, and 3D VLM is how the 3D scene is introduced, tailored to their specific scene representation formats, while all other parts remain consistent to ensure a fair comparison.

For the experiment in Table 2, which assesses the effectiveness of the world frame, the prompt templates in Figure 12 were used to evaluate model performance without a frame description.

Experiments in Tables 3 and 4 evaluate model performance in scenarios where the context change description is not given, with the exact prompt template provided in Figure 13.

---

### Reason without World Frame

Given a top-view of a 3D scene and a context change, imagine how the scene would appear after the change has been applied. Then, answer a question based on the modified scene.

Context Change: {}
Question: {}

The answer should be a single word or a short phrase.

The answer is:

---

*Figure 12.* Prompt template for evaluating model performance without using an anchor-based world frame.

---

### Reason without Context Change

Given a top-view of a 3D scene, mentally rotate the image to align with the specified orientation.

Scene Orientation: {}

Then, answer a question based on the aligned scene.

Question: {}

The answer should be a single word or a short phrase.

The answer is:

---

*Figure 13.* Prompt template for evaluating models in static scenes without context changes.

## B.3. More Quantitative Results

### B.3.1. MORE COMPLETE MAIN RESULTS.

We present the complete main results for each context change type and question type in Table 9. Additionally, we evaluate open-ended responses in the Hypo3D dataset using a GPT-based scoring approach, following the MSQA (Linghu et al., 2024) framework. Each GPT score $C$ is computed as:

$$C = \frac{1}{N} \sum_{i=1}^{N} \frac{s_i - 1}{4} \times 100$$

where $N$ is the number of questions, and $s_i \in [1, 5]$ is the discrete score assigned by GPT-4o-mini based on the question, ground truth, and model response (higher is better). The results are shown in Table 10.

Table 9. Complete EM and PM results of foundation models and human evaluators on Hypo3D.

| Input | Model | Metric | Movement | | | Removal | | | Attribute | | | Addition | | | Replacement | | | Overall |
|---|---|---|---|---|---|---|---|---|---|---|---|---|---|---|---|---|---|---|
| | | | Scale. | Dire. | Sem. | Scale. | Dire. | Sem. | Scale. | Dire. | Sem. | Scale. | Dire. | Sem. | Scale. | Dire. | Sem. | |
| Scene Captions | Llama 3.2 3B | EM | 29.62 | 15.89 | 30.94 | 35.29 | 14.14 | 30.54 | 26.50 | 13.49 | 32.23 | 30.26 | 12.49 | 32.12 | 26.20 | 10.96 | 33.08 | 26.08 |
| | | PM | 31.43 | 20.26 | 31.18 | 37.40 | 21.37 | 32.23 | 31.03 | 20.97 | 32.48 | 34.15 | 17.90 | 32.54 | 28.90 | 17.14 | 34.54 | 29.91 |
| | GPT-4o API (Text) | EM | 45.36 | 21.88 | 37.17 | 44.01 | 20.49 | 37.66 | 35.03 | 21.52 | 45.85 | 45.31 | 19.08 | 50.00 | 36.41 | 15.23 | 42.54 | 35.54 |
| | | PM | 47.20 | 26.01 | 37.17 | 46.59 | 29.67 | 40.32 | 39.43 | 30.74 | 47.54 | 48.65 | 24.44 | 50.84 | 39.55 | 20.85 | 44.65 | 39.65 |
| Semantic Top-View | Qwen2-VL 7B | EM | 34.82 | 19.96 | 35.49 | 33.73 | 17.27 | 35.76 | 29.14 | 15.25 | 40.53 | 35.50 | 14.49 | 37.99 | 30.29 | 13.00 | 41.54 | 29.85 |
| | | PM | 39.75 | 25.75 | 35.49 | 35.38 | 25.73 | 37.00 | 33.15 | 24.16 | 41.50 | 37.55 | 20.55 | 39.39 | 31.90 | 20.35 | 45.15 | 34.47 |
| | Qwen2-VL 72B | EM | 39.75 | 21.33 | 38.13 | 37.34 | 18.41 | 40.03 | 33.47 | 19.94 | 47.18 | 39.41 | 18.02 | 44.13 | 33.14 | 15.03 | 42.29 | 33.39 |
| | | PM | 42.18 | 27.06 | 38.13 | 38.71 | 26.72 | 41.09 | 36.54 | 27.14 | 47.79 | 43.11 | 23.97 | 44.41 | 34.19 | 22.29 | 43.66 | 37.51 |
| | LLaVA-OV 7B | EM | 36.35 | 20.34 | 36.93 | 32.61 | 16.13 | 38.29 | 33.24 | 20.29 | 39.53 | 38.75 | 13.31 | 40.50 | 31.76 | 14.06 | 39.80 | 30.62 |
| | | PM | 38.90 | 24.99 | 36.93 | 33.81 | 23.29 | 40.32 | 37.30 | 29.07 | 40.03 | 40.02 | 17.84 | 41.48 | 32.97 | 19.69 | 42.04 | 34.34 |
| | LLaVA-OV 72B | EM | 43.53 | 24.20 | 41.01 | 40.90 | 20.68 | 41.77 | 38.61 | 23.70 | 43.69 | 43.84 | 19.67 | 48.32 | 36.24 | 17.26 | 47.76 | 36.38 |
| | | PM | 44.73 | 29.38 | 41.01 | 42.17 | 29.13 | 42.88 | 42.43 | 32.38 | 44.24 | 45.10 | 25.03 | 48.32 | 37.78 | 24.43 | 49.13 | 40.13 |
| | Claude 3.5 Sonnet | EM | 21.28 | 8.65 | 26.62 | 18.64 | 11.48 | 29.27 | 26.17 | 12.90 | 26.91 | 25.76 | 7.54 | 33.24 | 24.33 | 7.37 | 26.87 | 20.42 |
| | | PM | 35.11 | 21.70 | 29.02 | 23.88 | 22.20 | 35.23 | 33.41 | 27.60 | 36.99 | 32.77 | 18.67 | 38.69 | 28.60 | 17.70 | 32.21 | 30.29 |
| | GPT-4o API | EM | 43.98 | 21.16 | 34.53 | 37.66 | 16.41 | 37.82 | 32.77 | 18.01 | 40.53 | 45.90 | 17.90 | 38.83 | 34.29 | 13.19 | 40.80 | 33.58 |
| | | PM | 45.08 | 25.24 | 34.53 | 38.71 | 24.08 | 38.82 | 35.77 | 25.02 | 40.97 | 47.52 | 21.97 | 39.39 | 35.62 | 18.66 | 42.04 | 36.75 |
| Non-Semantic Top-View | Qwen2-VL 7B | EM | 37.25 | 22.06 | 37.17 | 39.78 | 20.11 | 53.01 | 35.36 | 19.77 | 48.67 | 40.89 | 19.32 | 52.23 | 33.47 | 17.65 | 47.51 | 34.40 |
| | | PM | 40.83 | 28.04 | 37.17 | 41.58 | 28.25 | 53.80 | 38.74 | 27.76 | 50.11 | 42.71 | 25.80 | 54.47 | 35.32 | 24.31 | 52.36 | 38.91 |
| | Qwen2-VL 72B | EM | 45.81 | 25.05 | 46.52 | 50.50 | 27.13 | 63.13 | 47.29 | 29.74 | 68.27 | 51.81 | 24.85 | 64.80 | 47.76 | 26.96 | 63.43 | 44.25 |
| | | PM | 48.01 | 30.63 | 46.52 | 51.85 | 35.56 | 63.61 | 50.06 | 37.45 | 68.94 | 53.81 | 30.51 | 64.80 | 48.75 | 34.79 | 64.68 | 48.25 |
| | LLaVA-OV 7B | EM | 39.30 | 22.70 | 39.81 | 37.97 | 20.49 | 41.61 | 37.81 | 22.05 | 41.53 | 43.99 | 19.91 | 49.72 | 37.71 | 18.72 | 44.53 | 34.81 |
| | | PM | 41.04 | 27.44 | 39.81 | 39.09 | 28.27 | 43.04 | 41.22 | 30.11 | 43.05 | 44.88 | 25.62 | 51.96 | 38.57 | 26.22 | 47.51 | 38.60 |
| | LLaVA-OV 72B | EM | 46.93 | 27.11 | 47.00 | 48.69 | 26.94 | 51.42 | 48.33 | 29.97 | 56.48 | 51.22 | 25.68 | 57.26 | 49.80 | 27.06 | 56.22 | 43.01 |
| | | PM | 48.02 | 32.95 | 47.00 | 49.74 | 35.09 | 52.06 | 51.65 | 38.83 | 56.89 | 52.39 | 31.80 | 57.68 | 50.73 | 34.11 | 58.33 | 46.83 |
| | Claude 3.5 Sonnet | EM | 36.87 | 19.87 | 49.64 | 42.02 | 22.96 | 59.02 | 45.50 | 25.10 | 57.81 | 47.68 | 21.08 | 64.80 | 51.76 | 25.02 | 55.72 | 38.86 |
| | | PM | 48.39 | 32.67 | 50.60 | 47.64 | 34.49 | 65.11 | 51.01 | 39.29 | 71.23 | 52.64 | 34.28 | 69.27 | 54.22 | 35.97 | 61.82 | 48.65 |
| | GPT-4o API | EM | 49.51 | 25.52 | 51.32 | 52.18 | 26.47 | 61.71 | 50.21 | 29.15 | 68.95 | 58.52 | 27.44 | 61.73 | 50.29 | 25.41 | 60.70 | 45.50 |
| | | PM | 50.62 | 30.21 | 51.32 | 53.13 | 33.35 | 62.18 | 52.63 | 35.94 | 65.70 | 60.16 | 33.33 | 62.15 | 51.14 | 31.59 | 61.82 | 48.82 |
| Multi-View RGB-D | LLaVA-3D 7B | EM | 38.03 | 21.63 | 35.49 | 35.60 | 17.36 | 32.91 | 34.84 | 17.77 | 37.54 | 39.34 | 17.55 | 37.99 | 33.55 | 15.52 | 43.28 | 31.56 |
| | | PM | 39.38 | 26.86 | 35.49 | 36.74 | 25.04 | 33.57 | 38.72 | 26.22 | 38.18 | 40.44 | 24.03 | 38.13 | 34.88 | 22.34 | 44.15 | 35.23 |
| Point Cloud + RGB | LEO 7B | EM | 21.13 | 7.79 | 7.91 | 27.62 | 2.28 | 10.44 | 16.22 | 3.46 | 19.10 | 20.66 | 2.00 | 8.94 | 16.24 | 1.36 | 10.45 | 14.83 |
| | | PM | 30.05 | 14.54 | 10.07 | 31.60 | 5.80 | 17.01 | 24.91 | 7.60 | 26.99 | 32.12 | 10.66 | 13.83 | 23.70 | 8.63 | 15.67 | 22.40 |

Table 10. GPT Score of 10 foundation models and human evaluators on Hypo3D (Overall only). The best-performing model within each family is underlined.

| Model | GPT Score |
|---|---|
| *LLM (Scene Caption)* | |
| Llama3.2 3B | 28.13 |
| GPT-4o API (Text) | 37.89 |
| *2D VLM (Non-Semantic Top-View Map)* | |
| Qwen2-VL 7B | 32.01 |
| Qwen2-VL 72B | 35.58 |
| LLaVA-OV 7B | 32.29 |
| LLaVA-OV 72B | 38.20 |
| Claude 3.5 Sonnet API | 25.27 |
| GPT-4o API | 35.49 |
| *2D VLM (Semantic Top-View Map)* | |
| Qwen2-VL 7B | 36.74 |
| Qwen2-VL 72B | 45.90 |
| LLaVA-OV 7B | 36.91 |
| LLaVA-OV 72B | 45.11 |
| Claude 3.5 Sonnet API | 42.76 |
| GPT-4o API | **46.55** |
| *3D VLM (Multi-View RGB-D, Point Cloud)* | |
| LEO 7B | 17.47 |
| LLaVA-3D 7B | 33.80 |

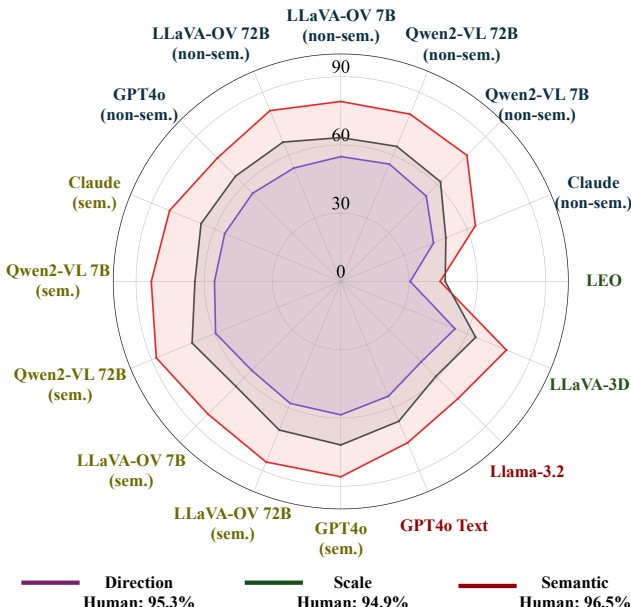

*Figure 14.* Model and human SBERT scores across question types. Models struggle the most with direction-based questions, followed by scale-based and semantic questions.

### B.3.2. MORE RESULTS ON DIRECTIONAL QUESTIONS

The radar chart in Figure 14 shows model performance across different question types using the SBERT metric, which measures cosine similarity between the text embeddings of predicted and ground-truth answers. These embeddings are generated using the SBERT model (Reimers, 2019). The results clearly show that most models struggle the most with directional questions, perform better on scale-based questions, and achieve their best performance on semantic questions.

Table 11 further highlights that, even when the top-view map is physically aligned with the world frame (i.e., no mental alignment required), model performance on direction-based questions shows no significant improvement, particularly for non-semantic maps. This indicates that current foundation models struggle with direction-based hypothetical reasoning, regardless of frame alignment.

*Table 11.* Comparison of model performance in direction-based questions on non-aligned versus aligned top-view maps. The results show no significant improvement, and in some cases, a decline in performance with aligned maps.

| Model | Non-aligned | | Aligned | |
|---|---|---|---|---|
| | EM | PM | EM | PM |
| *2D VLM (Non-Semantic Top-View Map)* | | | | |
| Qwen2-VL 7B | 16.70 | 23.93 | 16.60 | 23.40 |
| Qwen2-VL 72B | 19.21 | 25.95 | 19.99 | 27.08 |
| LLaVA-OV 7B | 17.91 | 24.08 | 18.86 | 25.29 |
| LLaVA-OV 72B | 21.97 | 28.81 | 21.69 | 28.25 |
| GPT-4o API | 17.95 | 23.64 | 18.64 | 24.69 |
| *2D VLM (Semantic Top-View Map)* | | | | |
| Qwen2-VL 7B | 20.22 | 27.18 | 21.94 | 29.17 |
| Qwen2-VL 72B | 26.77 | 33.65 | 33.95 | 43.28 |
| LLaVA-OV 7B | 21.28 | 27.82 | 23.77 | 31.64 |
| LLaVA-OV 72B | 27.60 | 34.74 | 31.93 | 40.77 |
| GPT-4o API | 26.57 | 32.67 | 32.38 | 42.53 |

*Table 12.* Performance comparison of models with and without Chain-of-Thought (CoT) prompting.

| Model | w/o CoT | w/ CoT |
|---|---|---|
| Llama3.2 3B | 23.91 | 26.08 |
| LLaVA-OV 72B | 42.78 | 43.01 |
| Qwen2-VL 72B | 44.90 | 44.25 |
| LLaVA-3D | 29.30 | 31.56 |

### B.3.3. EFFECT OF CHAIN-OF-THOUGHT

Figure 11 illustrates our use of the Chain-of-Thought (CoT) strategy, which explicitly decomposes the task into two steps: (1) imagining how the context change affects the scene, and (2) answering the question based on the altered scene.

To further examine the impact of CoT, we evaluated models using a simplified prompt structure:

```
Scene orientation:  {}
Context Change:  {}
Question:  {}
Answer:
```

The results in Table 12 show that removing CoT prompting leads to decreased performance in most models, except for Qwen2-VL 72B. This suggests that step-by-step reasoning supports hypothetical understanding to some extent. Nonetheless, model performance still falls short of human-level reasoning.

*Table 13.* Performance comparison of models with and without in-context learning (ICL).

| Model | w/o ICL | w/ ICL |
|---|---|---|
| Llama3.2 3B | 29.30 | 23.88 |
| LLaVA-OV 72B | 40.26 | 33.53 |
| Qwen2-VL 72B | 41.94 | 36.52 |

*Table 14.* Comparison of EM and PM scores for different models across Top and Multi-view settings.

| Model | View | EM | PM |
|---|---|---|---|
| LLaVA-OV 7B | Top | 34.81 | 38.60 |
| | Multi | 34.24 | 38.19 |
| LLaVA-OV 72B | Top | 43.01 | 46.83 |
| | Multi | 42.52 | 47.06 |
| Qwen2-VL 7B | Top | 34.40 | 38.91 |
| | Multi | 35.99 | 41.19 |
| Qwen2-VL 72B | Top | 44.25 | 48.25 |
| | Multi | 43.04 | 47.50 |

### B.3.4. EFFECT OF IN-CONTEXT LEARNING

We also investigated whether in-context learning (ICL) can enhance model performance in hypothetical reasoning tasks. Specifically, we applied three-shot ICL to both 2D VLMs and LLMs. The results in Table 13 show that ICL generally leads

Table 15. Effect of caption quantity on EM and PM scores to Llama3.2-3B.

| #Captions | EM | PM |
|---|---|---|
| 30 | 23.95 | 28.62 |
| 50 | 23.88 | 28.34 |
| 100 | 24.34 | 28.91 |
| 200 | 22.91 | 28.01 |

to a decrease in EM performance. This can be due to the limited number of examples failing to capture the diversity of context changes and question types in our dataset. Moreover, we observed that models often copied answers directly from the in-context examples rather than learning from them.

### B.3.5. EFFECT OF NUMBER OF VIEWS ON 2D VLMS

We evaluated 2D VLMs (semantic maps) using multi-view inputs (top, front, back, left, and right) compared to using top-view only. The results on 50 randomly sampled scenes in Table 14 show that performance remains comparable to using only the top view. This suggests that while multi-view inputs offer richer visual information, integrating visual features from different views presents another challenge for the models.

### B.3.6. EFFECT OF CAPTION DETAIL LEVEL ON LLMS

To assess how caption detail affects LLM performance on hypothetical 3D reasoning, we tested Llama3.2 3B with varying numbers of sampled captions. As shown in Table 15, more detailed inputs do not consistently improve performance, possibly due to the increased challenge of long-text reasoning. Following the SQA3D protocol, we use 30 randomly sampled object captions for the final scene description.

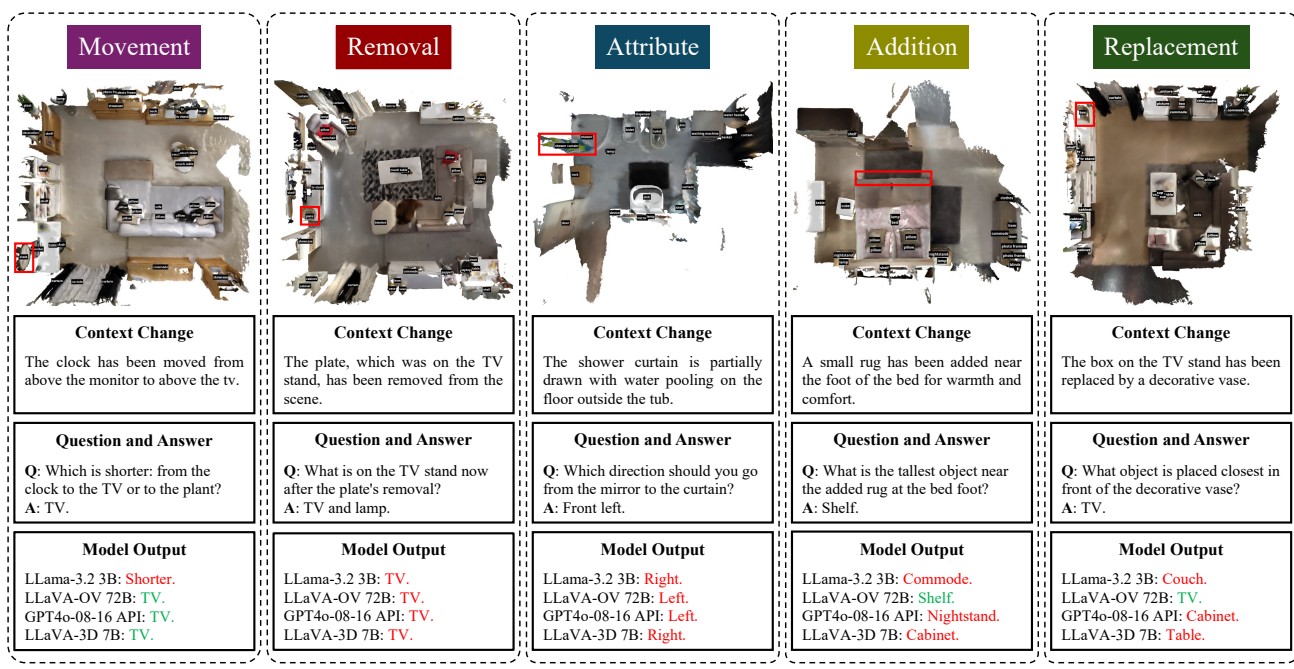

Figure 15. Qualitative Results. The changed object described in the context change is highlighted with a red bounding box. Model outputs are shown in green for correct and red for incorrect predictions. Results indicate that while models struggle with most examples, 2D VLMs are more likely to provide partially correct answers.

**B.4. More Qualitative Results**

The qualitative results of model performance across various context change types are shown in Figure 15. Although models answer most questions incorrectly, 2D VLMs (LLaVA-OV 72B, GPT-4o) are more likely to provide partially correct answers, suggesting a relatively better capability for hypothetical reasoning.

# C. Limitations and Future Work

One limitation is relying solely on text to describe context changes, which may lack precision for complex scenarios. Future work will incorporate multimodal approaches, such as images or egocentric videos, for more accurate and complementary representation. Additionally, our dataset focuses on hypothetical reasoning in indoor scenes, with plans to extend to outdoor environments. Lastly, Hypo3D addresses object-level changes (e.g., modifying specific objects); future work will explore scene-level changes involving significant layout rearrangements.

