# OpenReview forum: "Hypo3D: Exploring Hypothetical Reasoning in 3D"
_ICML.cc/2025/Conference — ICML 2025 poster_

### Official Review · Reviewer_aKvW · 2025-03-10

**Overall Recommendation:** 1

**Summary:**

This paper presents the construction of a new dataset, Hypo3D, for evaluating the hypothetical reasoning performance on 3D scene. Then, the authors conduct evaluation of several benchmark methods on this dataset and show huge performance gaps between human's performance and model's performance.

**Claims And Evidence:**

The authors claimed that existing models struggle to reason effectively in hypothetically changed scenes. Extensive experimental evaluation indeed validate this. But this claim does not involve any novel method. The only contribution is related to the dataset.

**Essential References Not Discussed:**

No.

**Experimental Designs Or Analyses:**

1) There is no proposed method, and the authors mainly evaluated existing approaches.
2) As discussed above, the authors put too much emphasis on LLMs, and overlook the importance of visual perception modules, yielding models' performance much lower than human's.

**Methods And Evaluation Criteria:**

1) There is no proposed method.
2) Evaluation criteria may be problematic. Most of the evaluation methods are based on large language models. To my understanding, in order to reason well, the perception module needs to be powerful enough to perceive as much useful information as possible, before performing any meaningful reasoning. The authors put too much emphasis on large language models while overlooking the capability of visual perception modules. As a result, the results are very poor compared to human performance.

**Other Comments Or Suggestions:**

No.

**Other Strengths And Weaknesses:**

1) The novelty in the dataset is limited. Conceptually, the dataset can be constructed by modifying existing 3D datasets through posing context changes, and revising the solutions of existing datasets in response to the context changes. (As we have full controls about the objects in the scene, this is convenient and easy.) As such, the value of constructing the Hypo3D dataset is greatly reduced.
2) The significance of this research is limited. No new method is proposed in this paper.
3) The so-called hypothetical reasoning is questionable. Before fully understanding the scene visually, hypothetical reasoning does not make too much sense. In addition, this problem in theory is similar to prediction-and-reasoning problem, i.e., given an existing scene, and the tendency of changing, we would like to reason over the future scene. Also, the context changing is provided in the form of text decription. This is difficult to achieve for real-world application where we have only a change in the scene while there is no such text description of context changes.

**Questions For Authors:**

In summary, I would like to see more justifications on:
1) The novelty of this paper.
2) The difference and significance of propose dataset in comparison to existing datasets.
3) More justifications on hypothetical reasoning over existing problem formulation.
4) More experimental results that utilize visual models to extract visual features for the scene, integrat with linguistic features, and leverage LLMs to better answer the questions.

**Relation To Broader Scientific Literature:**

The key contribution of this paper is the dataset. This dataset has limited value to the society.

**Theoretical Claims:**

There is no theoretical claim.

---

> ### Author Rebuttal · Authors · 2025-03-30
>
> Interesting.
>
> **Q1: About Novelty.**
>
> Our Hypo3D benchmark is a novel, methodological contribution in itself, as noted by other reviewers.  It focuses on structured evaluation and it is the first 3D reasoning benchmark with explicit question-type annotations, enabling fine-grained evaluation. Many influential multimodal reasoning benchmarks [1–3] focus on evaluation rather than proposing new models, yet remain highly impactful.
>
> Current vision models primarily handle basic spatial and semantic relationships but lack robust, holistic scene understanding. At the same time, reasoning in dynamic 3D environments remains a long-standing real-world challenge. We argue this challenge should not be deferred until all vision problems are solved. Instead, hypothetical reasoning offers a parallel research path, enabling progress on complex 3D reasoning alongside ongoing advances in foundational vision tasks. Furthermore, hypothetical reasoning is central to human cognition. As humans, we often reason with incomplete perceptual information, updating our mental models as new data emerges. This iterative process—forming, testing, and refining hypotheses—is fundamental to intelligent behavior. Integrating this capability into VLM is a step toward more human-like reasoning and a key milestone on the path to AGI.
>
> Our problem fits into the prediction-and-reasoning domain only when robust, real-time 3D scene updates are possible. However, current limitations in 3D reconstruction and editing, as noted in the paper, make accurate future scene prediction challenging. As a result, the task naturally shifts to an imagination (inference)-and-reasoning problem.
>
> Even widely used benchmarks like SQA3D could conceptually be constructed by posing situational descriptions and revising answers. Both our work and SQA3D deliberately avoided this. SQA3D emphasizes situational diversity, whereas Hypo3D focuses on diverse context changes. Our pipeline first collects a wide range of valid context changes and then generates corresponding questions and answers. Starting with fixed questions would have greatly limited feasible changes, especially given the strict constraints Hypo3D enforces on both changes and question design.
>
> Text descriptions may not always be available, but they remain a cost-effective and widely accessible alternative to reconstructing the full 3D scene after each change. Our future work will explore multimodal representation of changes  (e.g., images, depth) for more comprehensive hypothetical reasoning. Notably, in other reasoning tasks, early works like SQA3D relied on text to define the observer's situation, while later studies such as MSQA expanded to multimodal representations. This progression highlights the importance of text as a foundational step in developing new reasoning tasks.
>
> **Q2: Evaluation Criteria.**
>
> Most methods we evaluate are vision-language models (VLMs), not standalone LLMs. While VLMs include an LLM backbone, it is instruction-tuned with large-scale visual data (e.g., image captioning, VQA), endowing it with visual perceptual abilities beyond standard language understanding.
>
> We conducted additional experiments to examine the impact of visual encoders on VLMs' hypothetical reasoning. Specifically, we compared Cambrian-1 [4](using SigLIP, CLIP, DINOv2, and ConvNeXt) and LLaVA-Next (using only CLIP) [5], both sharing the same LLM backbone. Partial match results on semantic top-view maps below show that while leveraging more visual features can improve reasoning performance, a significant gap remains compared to human performance.
>
> | Model              | LLM Backbone     | PM     |
> |-|-|-|
> | Cambrian-1 13B     | Vicuna1.5-13B       | 41.80  |
> | LLaVA Next 13B     | Vicuna1.5-13B       | 38.87  |
> | Cambrian-1 34B     | Hermes2-Yi-34B       | 44.42  |
> | LLaVA Next 34B     | Hermes2-Yi-34B       | 41.47  |
> | Human     | -       | 92.50  |
>
> **Q3: Difference to Existing Problem Formulation.**
>
> Unlike traditional VQA tasks (2D or 3D), where answers can be directly inferred from the visual input, our hypothetical reasoning task introduces a key distinction: the visual input provides only prior context and is insufficient on its own.  Hypo3D shifts the focus from “see and answer” to the more complex cognitive process of “see, imagine, and answer”.
>
> [1] TopViewRS: Vision-Language Models as Top-View Spatial Reasoners, EMNLP, 2024.
>
> [2] SQA3D: Situated Question Answering in 3D Scenes, ICLR, 2023.
>
> [3] HourVideo: 1-Hour Video-Language Understanding, NeurIPS, 2024.
>
> [4] cambrian-1: a fully open, vision-centric exploration of multimodal llms, NeurIPS, 2024.
>
> [5] LLaVA-NeXT: Improved reasoning, OCR, and world knowledge, Arxiv, 2024

---

### Official Review · Reviewer_2kmQ · 2025-03-13

**Overall Recommendation:** 3

**Summary:**

The paper introduces Hypo3D, a benchmark task evaluating foundation models' ability to use hypothetical reasoning to "imagine" missing perceptual information in dynamic 3D scenes. It provides a dataset with various context changes and questions, showing that current models significantly underperform humans, frequently hallucinate irrelevant details, and struggle especially with movement changes and directional reasoning. The findings highlight current MLLMs' limitations in imagination-based reasoning, crucial for achieving human-like cognitive abilities.

## update after rebuttal
The rebuttal has addressed most of my concerns. The remained concern is that the key factors for the bottlenecks of 2D VLM and LLM in the hypothetical reasoning task need to be further discussed. Overall, the motivation for benchmarking MLLM's hypothetical reasoning ability is interesting. I will keep my rating.

**Claims And Evidence:**

The experiments on Hypo3D show the deficiency of current VLMs, supporting part of the motivation for proposing such a benchmark.

One concern is whether the input data types are appropriate for verifying the hypothetical reasoning ability of VLMs.

- For 2D VLM, the input data is a top-view image, which compresses the 3D scene in a plane. There exists information loss. Besides, the movement is a 3D transformation, which could not be imagined in the top view. The reason is more related to the 3D understanding ability of 2D VLM. Therefore, this type of data may not be appropriate for testing the hypothetical reasoning ability of 2D VLM.

- For LLM, the ability is related to the level of detail provided in the caption.

**Essential References Not Discussed:**

It is recommended that authors add a comparison between Hypo3D and current 3D grounding and captioning datasets, e.g. ScanRefer, MMScan, and a series of datasets.

**Experimental Designs Or Analyses:**

The experiments show current VLMs' ability of hypothetical reasoning on Hypo3D.

**Methods And Evaluation Criteria:**

The main concern is about the input data type problem mentioned before, which may not support the motivation of testing the hypothetical reasoning ability but the 3D imaginary ability.

**Other Comments Or Suggestions:**

Nil.

**Other Strengths And Weaknesses:**

The paper is well-written and organized.

**Questions For Authors:**

In fact, current 2D VLMs themselves may not be proficient at handling scale direction questions. The bad performance on Hypo3D comes from both hypothetical reasoning and fundamental ability. A good comparison experiment is testing the VLMs without adding changes in the scene, which can better help to find the key reason.

**Relation To Broader Scientific Literature:**

This paper proposes a benchmark to measure the ability of current VLMs' hypothetical reasoning in indoor scenes, which includes top-view images, scene caption, point cloud, and RGB-D video based on the ScanNet dataset. The benchmark demonstrates the deficiency of imagination.

**Theoretical Claims:**

There is no proof for theoretical claims.

---

> ### Author Rebuttal · Authors · 2025-03-30
>
> Thank you for your valuable feedback. We have added further explanations and additional experimental results to address your concern regarding the input data type issue.
>
> **Q1: 2D VLM Input**
>
> We adopted top-view images for 2D VLMs to maintain consistency with established 3D reasoning benchmarks like SQA3D. While top-view images cannot fully capture 3D geometry, they provide the most comprehensive spatial information in a single image.
>
> Besides, modern 2D VLMs have been able to extract depth cues from single images and reason about 3D positioning. This has been validated by our results, where 2D VLMs perform adequately on questions involving vertical reasoning (e.g., height). Also, Human evaluators also have reported that top-view images were generally sufficient for them to answer questions in our dataset.
>
> Additionally, we evaluated 2D VLMs (semantic maps) using multi-view inputs (top, front, back, left, and right) compared to using top-view only. The results on 50 randomly sampled scenes below shows that performance remains comparable to using only the top view. This suggests that while multi-view inputs offer richer visual information, integrating visual features from different views presents another challenge for the models.
>
> | Model         | View  | EM    | PM    |
> |---------------|-------|-------|-------|
> | LLaVA-OV 7B   | Top   | 34.81 | 38.60 |
> |               | Multi | 34.24 | 38.19 |
> | LLaVA-OV 72B  | Top   | 43.01 | 46.83 |
> |               | Multi | 42.52 | 47.06 |
> | Qwen2-VL 7B   | Top   | 34.40 | 38.91 |
> |               | Multi | 35.99 | 41.19 |
> | Qwen2-VL 72B  | Top   | 44.25 | 48.25 |
> |               | Multi | 43.04 | 47.50 |
>
> **Q2: LLM Input**
>
> To assess how caption detail affects LLM performance, we tested LLaMA-3.2 with varying numbers of sampled captions. As shown in the table below, more detailed inputs do not consistently improve performance—possibly due to the increased challenge of long-text reasoning. Following the SQA3D protocol, we use 30 randomly sampled object captions for the final scene description.
>
> | #Captions | EM    | PM    |
> |----------|-------|-------|
> | 30       | 23.95 | 28.62 |
> | 50       | 23.88 | 28.34 |
> | 100      | 24.34 | 28.91 |
> | 200      | 22.91 | 28.01 |
>
>
> **Q3: Dataset Comparison**
>
> The table below compares Hypo3D and existing 3D visual grounding (VG) and captioning datasets. Hypo3D is the first to annotate all question types and world frames, requiring models to understand hypothetical scenes.
>
> | Dataset             | Task                           | Question Type Annotation? | Hypothetical? | World Frame? | #Scans | #Language | Annotation         |
> |---------------------|--------------------------------|----------------------------|----------------|--------------|--------|-----------|---------------------|
> | ScanRefer           | VG                             | N/A                          | ✗              | ✗            | 0.7k   | 11k       | Human              |
> | Sr3D                | VG                             | N/A                          | ✗              | ✗            | 0.7k   | 115k      | Template           |
> | ScanQA              | QA                             | ✗                       | ✗              | ✗            | 0.8k   | 41k       | Template           |
> | SQA3D               | QA                             | ✗                        | ✗              | ✗            | 0.65k  | 33.4k     | Human              |
> | ScanScribe          | Captioning                     | N/A                        | ✗              | ✗            | 1.2k   | 278k      | GPT                |
> | MMScan              | VG + Captioning + QA           | ✗                          | ✗              | ✗            | 5.2k   | 6.9M      | GPT + Temp. + Human|
> | Hypo3D (Ours)       | QA                             | ✓                          | ✓              | ✓            | 0.7k   | 15k       | GPT + Human        |
>
> **Q4: Impact of Recognition on Reasoning**
>
> To mitigate the impact of object recognition on hypothetical reasoning, most experiments in the paper utilized semantic top-view maps with explicit text labels. We also evaluated VLMs on unchanged scenes in Table 3, where the performance was significantly higher. This confirms that the main challenge primarily lies in reasoning about hypothetical changes. All results above will be included in the final version of the paper.

---

> > ### Comment · Reviewer_2kmQ · 2025-04-09
> >
> > I appreciate the authors' exquisite rebuttal. The rebuttal addressed part of my concerns. There still are some concerns about the proposed benchmark.
> >
> > 1. There exists ambiguities for "Replacement Change" and "Addition Change". The context change don not provides the size of the newly added objects. It is not easy even for human to answer the scale and direction-related questions by imaging, when the new objects with huge size changes.
> >
> > 2. Existed 3d scene caption dataset do give the description of objects, relationships, coarse locations. However, it is hard to recover the accurate layout of the whole scene form these captions, since captions face a lot information losses, e.g. the accurate position of objects, the metric measurement of objects and relationship between objects. The LLM face challenges to fully understand the whole scene from these captions, so that it probably fails to finish the hypothetical reasoning tasks. This can be validated from Tab. 3 and 4 (LLaMA-3.2). The results in the rebuttal also indicate that no matter how much captions provided, the LLM fails to conduct the hypothetical reasoning. For 2D VLM input, the results indicates the similar conclusion in LLM. The 2D VLMs themselves fall short of understanding the whole from the top-view or multi-view images to conduct hypothetical reasoning.
> >
> > 3. The results on Tab. 4 and Tab. 3.
> > - Tab. 4 indicates that the irrelevant context change will leads to performance decline. And the similar decline happens after adding the relevant context changes for LLM and 2D VLM. Interestingly, for 3D VLM in Tab. 4, LLaVA-3D can adapt the irrelevant context changes.
> > - This phenomenon may indicate that 2D VLM and LLM fail to understand the context change phrase, since they face challenges to understand the whole scene. LLaVA-3D can understand the context change phrase, since it accepts 3D scene inputs and fully understand the whole scene. Therefore, the results of LLaVA-3D between Tab. 3 and 4 is a key observation, which can explain the LLaVA-3D is not good at hypothetical reasoning. However, the LLM and 2D VLM struggle for understanding the whole scene and the context change phrase not the hypothetical reasoning.
> >
> > 4. The insight of testing the hypothetical reasoning ability of existed models is interesting. It is important to find the real bottleneck of these models. It is better to entangle the key factors for the bottlenecks of 2D VLM and LLM.

---

> > > ### Author Response · Authors · 2025-04-09
> > >
> > > Dear reviewer,
> > >
> > > Thanks for your comment. Since the discussion period will be closed in half a hour, I can only address your concern via further explanation and cannot provide more experimental results.
> > >
> > > **1** For changes involving object replacements and additions, we observed ambiguity in the size of the newly added objects. As a result, our scale-related questions focus primarily on proximity (e.g., which object is closer to another) rather than comparing object sizes. Since the locations of added objects are precisely defined based on neighbor objects, the model can reliably answer proximity-based and direction-based questions. To further reduce ambiguity, we only ask about pairs of objects with clearly distinguishable locations. The only size-related questions we include for addition changes are of the form: "What object is the largest below the added object?" These questions do not require knowing the exact size of the added object. Instead, they focus on comparing the sizes of nearby objects relative to the added one, without involving the added object’s own size.
> > >
> > > **2**  We acknowledge that 2D  VLMs and LLMs do not have full access to the 3D scene, which limits their scene understanding capabilities. However, it is important to highlight that 2D VLMs currently achieve the best performance on hypothetical reasoning tasks. In this work, we closely follow the evaluation protocol from prior studies, selecting LLMs, 2D VLMs, and 3D VLMs to assess reasoning abilities. For LLMs, we provide ground-truth scene captions as input to maximize the accuracy of scene information conveyed through text. For 2D VLMs, which can only process image inputs, it is challenging to represent the full 3D scene using a single or even multi-view image. To address this, we use a semantic top-view map as input to reduce errors from object recognition and better capture the scene layout. While previous reasoning datasets often use non-semantic maps, we argue that our dataset optimizes input representation for 2D VLMs to the greatest extent possible.
> > >
> > > **3** We appreciate the reviewer's thoughtful analysis of Tables 3 and 4. However, we must respectfully clarify that the data doesn't support the conclusion that 2D VLMs primarily struggle with scene understanding compared to 3D VLMs. In fact, our results show that 2D VLMs consistently outperform LLaVA-3D across both tables. For example, Qwen2-VL 72B achieves 31.50% EM with changes in Table 3, significantly higher than LLaVA-3D's 20.50%.
> > >
> > > The key observation is that all models—regardless of architecture—show performance degradation when required to reason about hypothetical changes, though varying degrees. This suggests a fundamental limitation in hypothetical reasoning across current models rather than primarily a scene-understanding issue.
> > >
> > > **4** We fully agree that determining the real limitations of these models is crucial. However, as the reviewer is certainly aware, LLMs and VLMs are highly complex systems with numerous interconnected components, making it challenging to disentangle individual elements and examine them with perfect interpretability at a fine-grained level.
> > >
> > > Our current work represents an initial step toward understanding the hypothetical reasoning capabilities of existing models. In this first exploration, we prioritized evaluating these models as complete systems through our carefully constructed benchmark, which has already revealed significant performance gaps and interesting patterns across different model types and question categories.
> > >
> > > Moving forward, we plan to conduct a more detailed factor analysis that isolates specific bottlenecks for each model type. This analysis will broadly address two fundamental questions: (1) Can current models accurately perceive the scene and comprehend the context change? and (2) Assuming models can successfully perceive and understand the scene and context change, can they properly perform hypothetical reasoning?
> > >
> > > To achieve this goal, we will implement a more methodical, step-by-step evaluation approach that isolates components from the models, allowing us to better pinpoint the specific limitations in their hypothetical reasoning capabilities. This deeper analysis will provide more targeted insights for future model development.

---

### Official Review · Reviewer_RWSo · 2025-03-17

**Overall Recommendation:** 3

**Summary:**

The paper introduces a 3D reasoning benchmark called Hypothetical 3D Reasoning (Hypo3D). Specifically, the components of the benchmark can be summarized as follows. Consider a 3D scene representation (S) and a world frame from the scene (F) that contains an anchor object for specifying the direction to the model, e.g. “the table is to the left” will signal the model which direction is to the left. A set of context changes (C) are designed that are capture possible modifications to the scene S to obtain modified scenes S* (note that S* is not obtained). And finally, a set of question (Q) and answers (A) on the modified scene S*. The idea is to assess the “imagination” of existing foundation models to answer questions (Q) given the scene (S) and the context changes (C) by imagining what S* would look like. The authors consider three broad categories of questions: (1) scale-based, (2) direction-based, (3) semantic and include five types of context changes: (1) movement, (2) removal, (3) attribute, (4) addition, and (5) replacement. Experimental results for a variety of LLMs, 2D VLMs, and 3D VLMs are shown on this benchmark highlighting that the existing models fail in hypothetical reasoning.

## Update after rebuttal

I thank the authors for addressing all of my concerns and questions. I will keep my weak accept rating.

**Claims And Evidence:**

Most of the claims are clear. The major claim in the paper is that given the benchmark, existing LLMs and VLMs show unsatisfactory performance which is supported by the evaluation and experiments in the paper. However, I have a bunch of questions on the evaluation that are listed in the following sections.

**Essential References Not Discussed:**

I think the references are sufficient.

**Experimental Designs Or Analyses:**

**Benchmark design:**
* More insights in the groundtruth answers of the questions: in Line-240 (right column) it is mentioned that an answer contains 1.28 words on average. What do the answers look like? Are they always name of the objects, directions, numbers, etc.?
* How diverse are the directions defined using the world frame? What I mean by this is - from a given top-view image of a 3D scene that a person is looking at, let’s consider the person’s left as the “person-left” direction for the scene. But depending on the “left” that the world frame defines, that may or may not match with “person-left” -- if it matches how often it is? Or how often does it not match?
* Are there examples in which the anchor object is the object under consideration for the context change instruction? E.g. the same object that is used as anchor object is modified in the context change instruction.


**Evaluation:**
* While the use of exact match (EM) as a metric makes sense, I am not satisfied with the use of partial match (PM) as a metric that computes the percentage of overlapping words. The reason is because the answers on average are 1.28 words long, so given such short-length answers, PM doesn’t make sense. Also, is the overlap computed over words or tokens?
* Missing baselines: as the answers mostly would contain the name of objects, attribute of objects, and directions, it is very likely for the LLM/VLM to spit out random words by simply recognizing objects in the scene. So, one important baseline would be to ask the LLM/VLM to recognize the objects, their attributes, etc. in the scene (this will result in a set of words for a scene) and make a random prediction out of that set and then report the accuracy obtained.
* In-context learning: Was in-context learning tried? It would be great to see the performance of these models when in-context examples are provided.
* Chain-of-thought: Did the authors try CoT on Hypo3D? It would be also great to see the performance of these models using chain-of-thought prompting on this benchmark.
* Lines-359: “... most models exhibit severe hallucinations …” - any analysis on that? What do the model hallucinations look like? Are there any correlations with the question or context change type?
* How often does the model predict the anchor object as the answer?
* In”Insight 5”: do the evaluation consider questions in which there is no overlap of the object on which the question is asked and the object that is considered for context change. E.g. “what is the color of the coffee table” is the question and “move the couch to the left of the TV” is the context change.
* In Table 2, for the “w. Camera” results, the drop is accuracy wrt “w/o frame‘ is not much. Any further analysis on this?
* Any insights on why in Table 1: GPT4o (with non-semantic top-view map) performs worse than GPT4o (text only)?
* Lines-317-318 (left column): “... though it is not 100% due to the open-ended nature of Hypo3D questions … ” -- what does this mean exactly? Any examples?

**Methods And Evaluation Criteria:**

The paper proposes a benchmark by itself and uses that to study LLMs and VLMs. The design of the benchmark explores an interesting problem of “hypothetical reasoning” and is a good contribution to the community and makes sense to assess LLMs and VLMs.

**Other Comments Or Suggestions:**

N/A

**Other Strengths And Weaknesses:**

Most of my concerns are listed above. Although the benchmark is a great contribution, I feel the assessment of the LLMs and VLMs in the paper could have been more rigorous. E.g. evaluation using in-context learning, chain-of-thought, etc.

**Questions For Authors:**

I have included all of my concerns in the previous sections. I would like responses focussed on answering those concerns in the rebuttal.

**Relation To Broader Scientific Literature:**

I think the major contribution of the paper is the benchmark that can be broadly used to assess LLMs and VLMs to make them spatially aware and enable spatial reasoning. However, I think the evaluation process and metrics used in the paper need to be made more rigorous (as discussed above).

**Theoretical Claims:**

Not applicable.

---

> ### Author Rebuttal · Authors · 2025-03-30
>
> We sincerely appreciate your insightful feedback. We’re pleased to hear that you found our dataset to be a great contribution. All 2D VLM results reported below use the semantic map by default.
>
> **Q1: Answer Annotation**
>
> Similar to the answer types in Fig. 12 of SQA3D, our annotations include object names, attributes (e.g., shape, color, size, functionality, state, etc.), directions, numbers, and so on. The complete answer distribution will be included in the final version.
>
> **Q2: World Frame**
>
> We randomly rotated the top-view images before inputting them to VLMs, ensuring the world frame definition aligns with the person's orientation in 25% of scenes.
>
> **Q3: Anchor**
>
> 4.41% of context changes involve changes to the anchor object. E.g.,
>
> *Orientation: The shower is on the front side of the scene.*
>
> *Change: The shower has been moved to the left of the toilet.*
>
> **Q4: PM**
>
> Following TopviewRS, we used PM to measure word-level overlap, particularly for answers like "front" and "front left". We also employed SBERT scores to reduce evaluation bias in phrasing (Fig. 14). We further computed GPT-based score following MSQA, with results detailed in our response to **kxrB**. Both metrics show consistent ranking with PM.
>
> **Q5: Baseline**
>
> We conducted a baseline experiment with GPT-4o, which showed significantly lower EM accuracy than models in Table 1. This confirms that LLMs/VLMs are not merely recognizing objects or guessing, but possess limited capacity for hypothetical reasoning. Also, using semantic maps improved performance over non-semantic inputs, as VLMs benefit from explicit hover-text labels for more accurate recognition.
>
> |Model|Baseline|Reasoning|
> |-|-|-|
> |GPT-4o (non-sem.) |14.86|33.58
> |GPT-4o (sem.)|17.15|45.50
>
> **Q6: In-context learning (ICL)**
>
> We applied three-shot ICL to 2D VLMs and LLMs. ICL generally reduced EM performance, potentially because the limited example set cannot adequately represent the diversity in our context changes and questions. Moreover, we observed models directly copying answers from examples, indicating their inability to process long context effectively.
>
> |Model|w/o ICL|w/ ICL|
> |-|-|-|
> |LLaMA-3.2|29.30|23.88|
> |LLaVA-OV 72B|40.26|33.53|
> |Qwen2-VL 72B|41.94|36.52|
>
> **Q7: Chain-of-Thought (CoT)**
>
> We have utilized CoT to explicitly decompose the task into: (1) imagine how the change affects the scene, and (2) answer the question based on the changed scene.
>
> To further investigate CoT, we tested models with a simplified prompt:
>
> *Scene orientation: {}*
>
> *Context Change: {}*
>
> *Question: {}*
>
> *Answer:*
>
> Removing CoT prompting reduces performance in most models except Qwen2-VL 72B, suggesting that step-by-step reasoning aids hypothetical reasoning to some extent. Still, results lag behind human levels, and we plan to explore more advanced CoT methods in the future.
>
> |Model|w/o CoT|w/ CoT|
> |-|-|-|
> |LLaMA-3.2|23.91|26.08|
> |LLaVA-OV 72B|42.78|43.01|
> |Qwen2-VL 72B|44.90|44.25|
> |LLaVA-3D|29.30|31.56|
>
> **Q8: Hallucination**
>
> The hallucination in line 359 refers to the model incorrectly adjusting its answer in response to an irrelevant change, as noted in Insight 5. More examples of such hallucinations are provided in the response to Q10.
>
> **Q9: Anchor Object as Answer**
>
> The table displays the probability at which models predict anchor objects as answers. LLaVA-3D exhibits a much higher rate compared to both other models and ground truth frequencies, suggesting it tends to copy anchor objects as answers rather than engaging in reasoning.
>
> ||GT|GPT4o (Text)|GPT4o (sem.)|LLaVA-3D|
> |-|-|-|-|-|
> |Rate (%)|3.4|3.2|2.8|5.3|
>
> **Q10: No Overlap**
>
> We considered questions where the queried and changed object do not overlap. E.g.,
>
> *C: The lamp has been put onto the bath cabinet.*
>
> *Q: Where is the toilet paper relative to the trash can?*
>
> **Q11: World Frame Definition**
>
> The main difference between the w. camera and w/o frame settings is the inclusion of an additional camera view image. While models may not accurately interpret scene orientation from it, the image still provides extra visual context, so a significant accuracy drop is not expected.
>
> **Q12: Non-semantic vs. Text-only**
>
> GPT4o (non-sem.) underperforms due to difficulty recognizing objects in noisy scenes. In contrast, GPT (text-only) directly receives explicit object names and attributes from the caption.
>
> **Q13: Human Performance**
>
> Human performance falls short of 100% due to typos, formatting mismatches, and occasional misinterpretation of noisy scenes. Notably, less-than-perfect human performance is common in previous open-ended VQA benchmarks such as ScanQA and SQA3D. Below are human-failed examples:
>
> More examples can be found in our response to **kxrB**.
> |Question|Human|GT|
> |-|-|-|
> |How does the added Nescafe espresso machine's position compare to the paper towel?|Above the paper towel|Higher|
> |Which is closer to the added rubber gloves: the trash can or the paper towel roll?|Paper tower roll|Paper towel roll|

---

> > ### Comment · Reviewer_RWSo · 2025-04-08
> >
> > I thank the authors for addressing all of my concerns and questions. I will keep my weak accept rating.

---

> > > ### Author Response · Authors · 2025-04-08
> > >
> > > We are thrilled that our rebuttal and additional experiments have addressed all of your concerns. All results will be included in the final version of the paper.

---

### Official Review · Reviewer_kxrB · 2025-03-19

**Overall Recommendation:** 4

**Summary:**

This paper introduces a novel 3D-VQA benchmark called Hypo3D. Given a 3D scene representation (e.g., point clouds, BEV images) and a description of how the scene has changed, the model must infer the updated scene and answer a question based on it. The authors benchmark a range of open-source and closed-source 2D and 3D VLMs, identifying their limitations.

**Claims And Evidence:**

Most of the claims are supported.

**Essential References Not Discussed:**

NA

**Experimental Designs Or Analyses:**

Strengths
1. This work offers valuable insights into the performance of various models, which is much appreciated.

Weaknesses
1. For open-source 2D VLMs, the InternVL series is also widely recognized. It would be beneficial to include these models in the benchmark results.
2. Current 3D VLMs underperform, likely due to insufficient instruction-tuning for such tasks. To help the field better understand their limitations, it would be useful if the authors fine-tuned these models with a small amount of relevant data before conducting the benchmark.

**Methods And Evaluation Criteria:**

Strengths
1. This paper introduces a novel and compelling benchmark, tackling a crucial problem in 3D scene understanding—namely, how to handle changing scenes.
2. The authors provide a clear and detailed data collection process that involves human annotators, helping to ensure quality.

Weaknesses
1. According to Figure 2, the scene orientation descriptions rely on a top-view perspective (using terms such as “at the back of the scene” or “left side of the scene”). However, when assessing 3D-based VLMs, the inputs are point clouds and ego-view images rather than top-view maps, which may lead to ambiguity in interpreting orientation.
2. The benchmark primarily uses Exact Match (EM) and Partial Match (PM) for evaluation. While somewhat reasonable for single-word or short-phrase answers, these metrics may not reliably assess correctness if models produce longer responses. A potential improvement would be to employ GPT-4 to reformat the model outputs (e.g., summarizing them into a single word or short phrase) or to have GPT-4 judge correctness based on both the model’s response and the ground-truth answers. This approach could mitigate formatting-related biases and focus more accurately on the model’s scene-understanding capabilities.
3. Lines 315–318 reveal that human annotators do not achieve 100% accuracy on the benchmark, implying the presence of ambiguous samples. The authors may wish to analyze and remove such ambiguous cases before releasing the dataset, as this would be pivotal for the broader research community.

**Other Comments Or Suggestions:**

NA

**Other Strengths And Weaknesses:**

NA

**Questions For Authors:**

NA

**Relation To Broader Scientific Literature:**

Overall I think the proposed benchmark is novel and interesting.

**Theoretical Claims:**

NA

---

> ### Author Rebuttal · Authors · 2025-03-30
>
> We appreciate your insightful feedback. We’re glad to hear that you found our work to be novel and interesting.
>
> **Q1: Scene Orientation**
>
> For 3D VLMs using point clouds (e.g., LEO), inputs have been explicitly aligned to a top-view perspective with the floor on the XY-plane and vertical structures along the Z-axis. We acknowledge that the 3D VLMs taking ego-view images (e.g., LLaVA-3D) are expected to align  3D scenes internally for orientation understanding. Yet, they are provided with comprehensive information (i.e, multi-view RGB, depth, camera poses, and axis-alignment matrices) for implicit scene alignment. Again, while orientation may be more challenging to interpret for 3D VLMs compared to 2D VLMs, all models are provided with sufficient information to support effective orientation comprehension.
>
> **Q2: GPT-4 as Judger**
>
> Beyond EM and PM, we used SBERT (Fig. 14) for text similarity scoring to reduce evaluation bias in phrasing. Here, we explored GPT-based evaluation for open-ended responses following MSQA [1]. Each GPT score $\( C \)$ is formulated as:
>
>
> $$C = \frac{1}{N} \sum_{i=1}^{N} \frac{s_i - 1}{4} \times 100\%$$
>
> where $N$ is the number of questions,  $s_i \in [1, 5] $ (higher is better) is the discrete score assigned by GPT-4o-mini taking the question, ground truth, and model response as input. The scores for all models are shown below:
>
> | Model                    | GPT Score     |
> |-|-|
> | llama-3.2                | 28.13         |
> | GPT-4o (text-only)       | 37.89         |
> | Qwen2-VL-7B (non-sem.)   | 32.01         |
> | Qwen2-VL-72B (non-sem.)  | 35.58         |
> | llava-ov-7B (non-sem.)   | 32.29         |
> | llava-ov-72B (non-sem.)  | 38.20         |
> | Claude 3.5 Sonnet (non-sem.)        | 25.27         |
> | GPT-4o (non-sem.)        | 35.49         |
> | Qwen2-VL-7B (sem.)       | 36.74         |
> | Qwen2-VL-72B (sem.)      | 45.90         |
> | llava-ov-7B (sem.)       | 36.91         |
> | llava-ov-72B (sem.)      | 45.11         |
> | Claude  3.5 Sonnet (sem.)            | 42.76         |
> | GPT-4o (sem.)            | 46.55         |
> | LEO                      | 17.47         |
> | LLaVA-3D                 | 33.80         |
>
> Rankings of GPT scores closely align with PM rankings in Table 1, validating the reliability of PM.
>
> **Q3: Human Performance**
>
> Human performance rarely reaches 100% in open-ended VQA datasets. For example, the best EM@1 on ScanQA is 51.6%, and SQA3D reports 85–95% accuracy. Our dataset achieves even higher human scores, suggesting fewer ambiguities. Most errors are due to typos, vague phrasing, formatting mismatches, and inherent noise in 3D scenes. E.g.,
>
> | Question | Human Answer | GT |
> |-|-|-|
> | Where in the room would you stand to be the furthest from the liquid spill?                                 | Door                 | Next to the door       |
> | How does the placement position of the added Nescafé espresso machine compare with the paper towel?         | Above the paper towel| Higher                 |
> | Which is closer to the newly added rubber gloves, the trash can or the paper towel roll?                    | Paper tower roll     | Paper towel roll       |
>
> **Q4: InternVL-2.5**
>
> In response, we have evaluated InternVL-2.5 [2] (8B and 34B) on the Hypo3D benchmark using both semantic and non-semantic top-view maps. EM results show that InternVL-2.5 (8B version) consistently underperforms LLaVa-OV 7B and Qwen2-VL 7B across all settings on Hypo3D task.
>
> | Model                      | Movement | Removal | Attribute | Replacement | Addition | Overall |
> |---------------------------|----------|---------|-----------|-------------|----------|---------|
> | InternVL-2.5 8B (non-sem.)     | 20.07    | 19.97   | 19.22     | 16.12       | 20.57    | 19.38   |
> | InternVL-2.5 38B (non-sem.)    | 26.56    | 27.67   | 27.02     | 24.47       | 29.47    | 27.01   |
> | InternVL-2.5 8B (sem.)         | 21.96    | 24.82   | 27.41     | 16.92       | 27.74    | 23.83   |
> | InternVL-2.5 38B (sem.)        | 30.66    | 36.77   | 37.42     | 35.36       | 38.37    | 35.18   |
>
> **Q5: Instruction-tuning**
>
> We agree that exploring instruction-tuned 2D and 3D VLMs on our task would be valuable. Given the limited availability of computing resources, we intend to carry out a systematic study as part of our future work. We currently focus on zero-shot evaluation to ensure a fair comparison of all models. Notably, LLaVa-3D (7B), the SoTA 3D VLM, performs comparably to similar-sized 2D VLMs, suggesting that 3D VLMs' relatively weaker performance may stem from smaller model sizes.
>
> [1] msqa: multi-modal situated reasoning in 3d scenes, NeurIPS, 2024.
>
> [2] Expanding performance boundaries of open-source multimodal models with model, arxiv, 2024.

---

> > ### Comment · Reviewer_kxrB · 2025-04-05
> >
> > I appreciate the authors' rebuttal and think the contents in the rebuttal should be included in the final version of the paper.

---

> > > ### Author Response · Authors · 2025-04-06
> > >
> > > We sincerely appreciate your review of our rebuttal. We will certainly add all the results presented in the rebuttal in the final version of the paper. If you have any further questions regarding our work, please feel free to contact us.

---

### Decision · Program_Chairs · 2025-05-01

**Decision:**

Accept (poster)

**Comment:**

This paper was reviewed by four experts in the field, and received three positive reviews and one negative (4, 3, 3, 1). The positive reviews commended the utility of the proposed dataset (with one reviewer calling it novel, compelling, crucial), and raised some minor clarification questions, which the authors diligently addressed in a well-crafted rebuttal. The negative review complains that the paper does not propose a new method, but ICML guidelines allow novel datasets to be contributed too. The negative review further states that the dataset/benchmark is easy to create, but the authors dispute this convincingly. The AC's recommendation is to accept.